


# Compound flood modelling framework for rainfall-groundwater interactions

Francisco Peña[12345], Fernando Nardi[13], Assefa Melesse[4], Jayantha Obeysekera[5], Fabio Castelli[2], René M. Price[34], Todd Crowl[3], Noemi Gonzalez-Ramirez[6]

[1]WARREDOC, University for Foreigners of Perugia, Perugia, 06123, Italy
[2]Department of Civil and Environmental Engineering (DICEA), University of Florence, Florence, 50139, Italy
[3]Institute of Environment (InWE), Florida International University, Miami, FL, 33199, USA
[4]Department of Earth and Environment, Florida International University, Miami, FL, 33199, USA
[5]Sea Level Solutions Center, Florida International University, Miami, FL, 33181, USA
[6]Riada Engineering, Inc. P.O. Box 104, Nutrioso, AZ 85932, USA

*Correspondence to*: Francisco Peña (fpena023@fiu.edu)

**Abstract.** Compound floods are an active area of research where the complex interaction between pluvial, fluvial, coastal or groundwater flooding are analyzed. A number of studies have simulated the compound flooding impacts of precipitation, river discharge and storm surge variables with different numerical models and linking techniques. However, groundwater flooding is often neglected in flood risk assessments due to its sporadic frequency - as most regions have water tables sufficiently low that do not exacerbate flooding conditions -, isolated impacts and considerably less severity in respect to other types of flooding. This paper presents a physically-based, loosely-coupled modelling framework using FLO-2D and MODFLOW-2005 that is capable to simulate surface-subsurface water interactions to represent compound flooding events in North Miami. FLO-2D, responsible of the surface hydrology and infiltration processes, transfers the infiltration volume as recharge to MODFLOW-2005 until the soil absorption capacity is exceeded, while MODFLOW-2005 return exchange flow to the surface when groundwater heads are higher than the surface depth. The model calibration is based on three short-lived storm events that as individual processes represent minimum flooding conditions but in combination with pre-existing high-water table levels results in widespread flooding across the study area. Understanding groundwater flood risk is of particular interest to low-elevation coastal karst environments as the sudden emergence of the water table at ground surface can result in social disruption, adverse effects to essential services and damage infrastructure. Results are validated using FEMA's severe repetitive loss (SRL) property records and crowdsourced data. Further research should assess the exacerbated impacts of high tides and sea level rise on water tables under current and future climate projections.



## 1 Introduction

Flood inundation modelling is of critical importance for better planning, forecasting and decision-making practices (Teng et al., 2017). Scientific and technological innovations in numerical algorithms have continuously improved the performance of physically-based hydrologic, ocean circulation and hydraulic modelling packages to simulate faster and more accurate flood

physical processes over the computational domain at various scales and resolutions (Devia et al., 2015). However, most flood inundation models are designed to simulate specific flood hazards (i.e., pluvial, fluvial, coastal, groundwater) independently and are unable to assess complex flood dynamics per se due to code limitations and burdensome compatibility. To address these numerical constraints, some models have the ability to operate as linked units or groups by using coupling schemes (i.e. one-way, loosely, tightly, fully) to build compound models capable of simulating multiple flood drivers (Santiago-Collazo et

al., 2019).

Compound floods (CF) are high-impact low-probability events characterized by a non-linearity behavior resulted from the complex interactions of interrelated flood drivers triggered at the same spatial and temporal scales (Field et al., 2012; Seneviratne et al., 2012; van Westen & Greiving, 2017; Zscheischler et al., 2018). Research on CF field has received increasing

attention in recent years due to their adverse impacts at the global scale. Deterministic and probabilistic approaches are preferred frameworks to analyze CF events. Stochastic models through copula-based probability analysis and extreme value theory examine the interrelationship between flood drivers, while physically-based numerical simulations provide a tangible depiction of the flood dynamics for current and future climate projections. Several compound flooding studies have used physically-based hydrodynamic models as the reference model to simulate the combined effects of rainfall-runoff and storm

surge (Christian et al., 2015; Ikeuchi et al., 2017; Karamouz et al., 2015; Kumbier et al., 2018; Olbert et al., 2017). Failure to consider the compound interactions of flood drivers can result in significant uncertainties in the magnitude, timing, and estimation of flood risk (Wahl et al., 2015). Therefore, the transition from traditional univariate approaches to a multivariate perspective is necessary to improve flood hazard understanding and predictions (Bates et al., 2021).

The significance of groundwater flooding is rarely disputed as it is only relevant to geographical regions sitting on top of permeable rock that are prone to groundwater emergence (i.e., Miami, Yucatán Peninsula, United Kingdom). Groundwater floods are events limited to prolonged rainfall in low-elevation karst watersheds characterized by unconfined aquifers that experience sudden increases of already high-water table levels above normal conditions (Finch et al. 2004). Although there has been a substantial increase in groundwater flooding literature since the 2000s as well as advances in understanding surface

water/groundwater interactions (Brunner et al., 2017; Sophocleous, 2002), relevant knowledge gaps and lack of understanding of this phenomenon persist from the complex relationship between topography and hydrogeology (Bradford, 2002; Hughes et al., 2011; Ó Dochartaigh et al., 2019). The water table response time to hydrological events is controlled by the soil, vegetation and aquifer properties, which influence the infiltration capacity, recharge rate and response time (Nalesso, 2009). Similarly,



the groundwater dynamics are influenced by spatial-temporal variations of single or compound flood drivers (i.e. precipitation
events, high river levels, above-average tides and sea level rise conditions) over long or repetitive periods of time (Ascott et
al., 2017). Thus, the water table response to hydrological mechanisms (García-Gil et al., 2015), system fluctuations and
residence time (MacDonald et al., 2014) determine the severity of groundwater flooding.

While probabilistic and empirical approaches have contributed to the development of regional groundwater flood maps (Cobby
et al., 2009; Jacobs, 2007), physically-based models are scarce. Abboud et al. (2018) found that the June 2013 compound flood
disaster in the Elbow River (Canada) was induced by steady precipitation and increased river flow discharges from upstream
basins resulting in basement flooding due to the rise of the water table. The combined effects of fluvial and groundwater
flooding were not considered on that study since the MODFLOW river package focused exclusively on groundwater flow.
Similarly, Yu et al. (2019) applied the coupled surface-subsurface model PIHM to produce a comprehensive groundwater
flood risk and damage assessment over the Koiliaris River (Greece). Yang & Tsai (2020) investigated the impacts of water
table dynamics on groundwater flooding and levee under seepage in New Orleans, Louisiana using MODFLOW-USG for
hazard mapping, flood delineation and levee breach analysis. Su et al. (2020) developed a coupled model to assess the improved
response of the repaired storm drain system infrastructure with the shallow aquifer groundwater dynamics by coupling EPA
SWMM with MODFLOW-2005 at the city of Hoboken in New Jersey (USA).


Previous efforts to model groundwater levels in South Florida have been developed in the form of hydrogeologic maps (Fish
& Stewart, 1991), estimation of aquifer parameters to calculate groundwater flow (Cunningham et al., 2004), and statistical
analysis of hydrological measurements (Chebud & Melesse, 2011, 2012; Prinos & Dixon, 2016). Similarly, Hughes and White
(2016) investigated the effect of pump practices and sea level rise on surface water routing and groundwater interactions in
MDC using MODFLOW. Currently this is the main reference model for MDC regional research and planning purposes in
hydrologic, ecologic, and environmental fields. Regarding the study area, Sukop et al. (2018) developed a MODFLOW model
that analyzed the current and future response of the water table to rainfall events in a portion of the Arch Creek Basin. The
study highlighted precipitation as the main trigger for groundwater-induced flooding, with tidal fluctuations and sea level rise
increasing the shallow water table. Researching the flood risk potential from surface-subsurface water interactions in MDC
where the water table is near to the ground surface is critical as it could reveal hidden risks from the compound impact of
major storms and coastal forcing variables for present and future scenarios.

The main purpose of this study is to develop a loosely-coupled modeling framework capable of addressing complex compound
flood phenomena. To better understand the combined effects of pluvial and groundwater flooding in a low elevation coastal
zone, a methodology is developed to couple the 2D hydrodynamic software FLO-2D and the groundwater model MODFLOW-
2005. The Arch Creek Basin in North Miami was selected as an ideal test site due to its unique hydrogeomorphology, low-
lying topography, influence of tides on drainage outlets, and high vulnerability to flooding events. Three events characterized





by short-lived heavy precipitation events and unusually high-water tables were selected to simulate the surface groundwater interaction. Finally, the coupled model results were validated based on official reports and volunteered geographic information (VGI) flood observations from the study area. This study aims to highlight the importance of groundwater flooding as a potential flood driver in urbanized karst coasts. The paper is organized as follows: a complete description of the study area is introduced (Section 2), followed by data collection and the methodology presented in Sections 3 and 4. Results illustrate the main findings (Section 5); the discussion compares the results with similar work in the region (Section 6); and the conclusion section including the advantages, limitations, and future research (Section 7).

## 2 Study Area

### 2.1 Site description

The Arch Creek Basin is located in the northeastern part of Miami-Dade County (MDC), along the coast of Biscayne Bay in the city of North Miami, Florida. Prior to anthropogenic interventions, the Arch Creek River served as an important flow corridor that connected the Everglades to Biscayne Bay, controlling the flood pulse dynamics in the tropical wetland system (Fig. 1).

The gradual modifications in land use and the construction of the Biscayne Canal in the 1920s marked the transition of the natural environment to agricultural lands. Variations in the soil moisture conditions and infiltration levels due to changes in the streamflow and drainage patterns in the area caused unsustainable farming practices that lead to a shift to residential development (Fig. 2). The urbanization process along Biscayne Bay required considerable cut and fill earthworks to create ideal urban development conditions (Miami-Dade, 2016).

The Arch Creek Basin (16.95 km$^2$) is a low-lying coastal zone predominantly urbanized (90.1%) and economically diverse. The population is distributed within five jurisdictions, primarily concentrated in North Miami and North Miami Beach (Table 1). Although the topography is predominantly low and flat, some areas within the basin are considered the highest elevations in MDC ranging from 5 to 15 meters.

Frameworks to integrate flood risk mitigation and climate change adaptation strategies are a main component in Miami Dade County's policy agenda (GM&B, 2019). As a result, the Arch Creek Basin received the designated status of "Adaptation Action Area", the first pilot project in Florida to build social, environmental and economic resilience (Miami-Dade, 2016).





## 2.2 Climate

The climate of southeast Florida is characterized by wet (May to October) and dry seasons (November through April) with 75% of the annual rainfall occurring in the wet season (Abiy et al., 2019). The average annual rainfall in Miami is above 1500

mm and the average monthly precipitation during the wet season is above 150 mm (Abiy et al., 2019). Rainfall can vary from year to year (1000 – 2000 mm/yr), due to tropical storms and extreme hydrometeorological events which highly influence rainfall amounts. A reported increasing trend in rainfall of 2.1 mm/yr from 1906 to 2016, mainly attributable to an increase in wet season rainfall (Abiy et al., 2019), underscores that south Florida is under a continued threat from flooding.

## 2.3 Hydrogeology and groundwater

The Arch Creek Basin sits atop one of the most permeable aquifers in the world, known as the Biscayne Aquifer. The Biscayne Aquifer stores 34 billion m$^3$ of water and spans an area of 10,000 km$^2$ (Price et al., 2020) tapering from near the center of peninsula Florida towards the eastern coastline where its maximum thickness is about 38 meters (Parker & Cooke, 1944) and hydraulic conductivities exceeding 3,000 m/day (Fish & Stewart, 1991).

The stratigraphy of Biscayne aquifer consists entirely of unconfined permeable limestones of the Fort Thompson and Miami Limestone Formations and contains numerous solution conduits, resulting in rapid infiltration and recharge to the aquifer (Cunningham & Florea, 2009; Hoffmeister et al., 1967; Parker & Cooke, 1944). Recharge via precipitation occurs primarily in the Everglades and groundwater flows eastward towards the shore where it discharges to Biscayne Bay (Cunningham & Florea, 2009).

**3 Data description**

This section presents the data sets required to build the 2D surface-subsurface flood modelling study, including the topographic input, and hydrologic monitoring stations that provide rainfall, tide and well gauge records, as well as verified flood observations.

## 3.1 Topography

The Light Detection and Ranging (LiDAR) digital elevation model (DEM) is a 2-meter spatial resolution produced by Miami-Dade County, Florida. The LiDAR scanner corresponds to the actual bare-earth surface, removing tops of vegetation, buildings, and vehicles, and the project coordinate system is UTM zone 17N Horizontal Datum WGS84. In terms of elevation, the North American Vertical Datum of 1988 (NAVD 88) was assigned as the reference geodetic vertical datum for this study, substituting the original measurements based on the National Geodetic Vertical Datum of 1929 (NGVD 29).





## 3.2 Hydrologic input

Hydrologic modeling included hydrologic conditions of the time periods 1-4 October 2000, 6-8 June 2013, and 23-26 May 2020. Boundary and initial hydrologic inputs such as precipitation, tide and ocean-side water levels, and groundwater heads over the specified time periods were obtained from the following sources.

### 3.2.1 Rainfall

The NEXRAD Radar Rainfall Application is a scientific web map interface developed by the South Florida Water Management District (SFWMD) on which rainfall data is reported based on spatial coverage configurations in the form of the entire district, counties, Arch Hydro Enhanced Database (AHED) watersheds, or Rain Grid. The NEXRAD Rain Grid Layer is a 2 km grid resolution that provides an accurate representation of precipitation every 15 minutes. Rainfall Grid cell 10044042 was selected to characterize the Arch Creek Basin's rainfall conditions.

### 3.2.2 Tides and ocean-side water levels

DBHYDRO is the official SFWMD repository for climate, hydrologic, and environmental databases (https://www.sfwmd.gov/science-data/dbhydro). Ocean-side water levels were obtained from stations S28_H and S28_T, located in the Biscayne Canal Number C-8 on the Arch Creek southern boundary edge.

The NOAA Tides & Currents website (https://tidesandcurrents.noaa.gov/) provides local water levels, tides, current predictions, and other oceanographic and meteorological conditions. The closest coastal sensor to the Arch Creek Basin is located at the Virginia Key, Biscayne Bay Station (ID #8723214).

### 3.2.3 Groundwater heads

The Unites States Geological Survey (USGS) National Water Information System (https://waterdata.usgs.gov/nwis/gw), in cooperation with the SFWMD, records daily summary data of maximum groundwater levels in the south Florida region. The groundwater level data was obtained from well G-852 adjacent to the outer western boundary of the study area (Fig. 3). Daily field water level measurements have been recorded since 1973, and 15-minute intervals since October 2007.

## 3.3 Repetitive flood claims

FEMA's severe repetitive loss (SRL) properties program is designed to provide grants and financial assistance to residential properties that have experienced frequent flood losses over the years (FEMA, 2021). Currently, seventy-five properties have requested financial assistance for property acquisition or to recoup with some of their investments due to flood damages in the





Arch Creek Basin (Miami-Dade, 2017). The database stores detailed information on the date of loss, building type, flood zone
designation, type of insurance and claim payments between 1995 to 2015, providing a clear footprint of flooding risk hotspots
and flood prone communities.

## 4 Methodology

### 4.1 Hydraulic Model: FLO-2D

FLO-2D is a physically-based volume conservation model capable of simulating the propagation of water dynamics over
confined and unconfined environments using the dynamic wave approximation to the momentum equation (O'Brien et al.,
1993). The flood routing model combines hydrology and hydraulics in a computational square grid system environment that
moves the flood volume across the tile's boundary one step at a time. Rainfall-runoff processes can come in the form of rainfall
data over the domain or distributed input flood hydrographs in the channel or floodplain. 1D equations are applied for channel
flow routing movement in a downstream direction as long as the flow remains in the channel cross-section. Conversely, 2D
equations are activated when the maximum capacity of the channel is exceeded, and during overland runoff in the floodplain.

The model can represent high-resolution environments and urban features, including buildings, streets, levees, obstructions,
and drainage systems. These can influence the flow distribution dynamics, which are governed by the topography and Manning
roughness coefficient. Similarly, the flow propagation and velocity can be influenced by abrupt changes in slope, depressions,
unsteady flow conditions, and hydraulic structures.

The FLO-2D model input data are the floodplain topographic digital terrain model (DTM), channel geometry, inflow and
outflow boundary conditions, as well as grid cell parameters representing the presence of artificial features on the bare earth
(i.e., levees, building, bridges) (O'Brien, 2011).

The equations implemented in the model consist of the Continuity Equation:

$$\frac{\partial h}{\partial t} + \frac{\partial hV}{\partial x} = i \tag{1}$$


and the Momentum Equation:

$$S_f = S_o - \frac{\partial h}{\partial x} - \frac{V}{g} \cdot \frac{\partial V}{\partial x} - \frac{1}{g} \cdot \frac{\partial V}{\partial t} \tag{3}$$





where $h$ is the flow depth, $t$ is the time variable, $V$ is the depth-averaged velocity in one of the potential eight flow directions $x$, $i$ is the excess rainfall intensity (if the rainfall component is considered), $So$ is the bed slope, $g$ is the gravity acceleration, and $Sf$ is the friction slope based on the Manning Equation. For the Momentum Equation, the bed slope is subtracted by the pressure gradient, local, and convective acceleration variables respectively, to represent the one-dimensional depth-averaged channel flow.

FLO-2D uses the abovementioned equations of motion to calculate the average flow velocity across a grid element boundary one direction at a time in eight potential flow directions over the floodplain, four cardinal directions (North, East, South, and West), and four diagonal directions (NE, NW, SE, and SW). The calculation of each velocity is one dimensional and solved independently from the other boundary cells; thus, velocity vectors are not calculated when the flow is shared with adjacent grid cells. The stability of this explicit numerical scheme is based on strict criteria to control the magnitude of the variable computational time-step.

### 4.2 MODFLOW-2005

MODFLOW is the world's leading open-source groundwater flow model used by hydrologists. MODFLOW is a computer code developed by the USGS since 1984 that uses Fortran language to simulate groundwater flow aquifer layers (confined or unconfined) using a block-centered finite-difference approach (Harbaugh, 2005). Technological developments have contributed to overall updates in the code, resulting in the much-improved version of MODFLOW-2005. MODFLOW-2005 processes are structured as flow packages, which are divided into multiple subroutines that are responsible for simulating optional processes that deal with a single aspect of the simulation, including block-centered flow (BCF6), layer property flow (LPF), unsaturated zone flow (UZF), and seawater intrusion (SWI2) to mention a few. Similarly, the model offers several solvers to solve matrix equations, as well as subsidence, observations, surface-water routing, and transport packages.

The geometric discretization of the aquifer(s) is fundamental to transform the aquifer components into discrete elements. The aquifer is broken down in grid elements to obtain the number of rows, the number of columns, and the width of each row and column for the horizontal direction. The vertical water pressure direction is delineated in the model by specifying the number of layers to be used, and the top/bottom elevations of every cell and layer. The number of layers corresponds to the number of aquifers. The spatial grid resolution must be appropriate to the domain and scale to set the model boundary conditions, as well as the aquifer characterization and parameters in specific cells, to represent with the highest standard of accuracy the modeling components for surface-subsurface flow interactions. At the end of the simulation, all cell centroids (also known as nodes) will record the flow stresses of the hydrogeological system, such as water heads, recharge, and zetas.

The following expression illustrates the three-dimensional groundwater movement at constant density through porous earth material using MODFLOW:



$$\frac{\partial}{\partial x}\left(K_{xx}\frac{\partial h}{\partial x}\right) + \frac{\partial}{\partial y}\left(K_{yy}\frac{\partial h}{\partial y}\right) + \frac{\partial}{\partial z}\left(K_{zz}\frac{\partial h}{\partial z}\right) + W = S_S\frac{\partial h}{\partial t} \tag{3}$$

where $K_{xx}$, $K_{yy}$, and $K_{zz}$ are values of hydraulic conductivity along the $x$, $y$, and $z$ coordinate axes, which are assumed to be parallel to the principal axes of hydraulic conductivity, $h$ is the potentiometric head, $W$ is a volumetric flux per unit volume representing sources or sinks of water, with $W < 0.0$ for flow out of the groundwater system, $W > 0.0$ for flow into the system, $S_S$ is the specific storage of the porous material, and $t$ is time.

The presented groundwater flow equation follows the application of the continuity equation, preserving the balance of flow between inputs-outputs with changes in the storage capacity. Under the premise that water density remains constant, the continuity equation expressing the balance of flow for a cell is calculated as:

$$\sum Q_i = S_S\frac{\Delta h}{\Delta t}\Delta V \tag{4}$$

where $Q_i$ is the flow rate into the cell, $S_S$ is the specific storage or the volume of water that can be injected per unit volume of aquifer material per unit change in head, $\Delta V$ is the volume of the cell, and $\Delta h$ is the change in head over a time interval of length $\Delta t$.

**4.3 Coupling surface-groundwater models**

FLO-2D is capable of simulating coupled hydrodynamic interactions of surface and subsurface flow components with
MODFLOW-2005. A loosely-coupled technique interaction approach was applied to combine FLO-2D, and MODFLOW-2005 Groundwater Flow Process (GWF) package input to solve the flood routine and groundwater flow numerical equations separately by exchanging information in an iterative matter (Santiago-Collazo et al., 2019) (Fig. 4).

The resulted surface-subsurface water exchanges under unsteady flow can occur at any given time of the simulation in the
discretized domain, as groundwater recharge from water infiltration in floodplains and rivers or as groundwater return exchange flow when the water table reaches the surface.

The main factors determining the coupling compatibility process between FLO-2D and MODFLOW-2005 include the algorithms' mathematical solver compatibility to calculate and transfer the exchanged volumes in opposite directions and share
consistent spatial and temporal scales. A significant advantage in the coupling process is that both numerical codes are written in FORTRAN programming language and shared the same explicit finite difference method. Thus, the spatial and temporal intervals of FLO-2D and MODFLOW-2005 are separated into a selected number of time steps, and the solution is calculated by solving the two- and three-dimensional equations, respectively. From a numerical perspective, this independence is beneficial to satisfy the numerical stability criteria and accuracy.






In terms of the spatial scale, a perfect match between FLO-2D and MODFLOW-2005 surface elevation layers is necessary for the surface and subsurface water interactions to happen. This agreement is subject to identical geographical position, reference system, size resolution, and topographic cell elevations (Fig. 5). Although the coupled models can have variations in the number of cells and domains, FLO-2D cells must overlap the MODFLOW-2005 grid domain system to compute results and

transfer the output data from one model to another and vice versa until the end of the simulation.

It is important to note that FLO-2D and MODFLOW-2005 design structures present significant operability differences to perform calculations. In MODFLOW-2005, the simulation is divided into a series of stress periods within which specified data

are constant. Each stress period, in turn, is divided into a series of time steps. The solution of the finite difference equations can be written in matrix form as:

$$[A]\{h\} = \{q\} \tag{5}$$

where $[A]$ is a matrix of the coefficients of the head for all active nodes in the grid, $\{h\}$ is a vector of head values at the end of time step $n$ for all grid nodes, and $\{q\}$ is a vector of the constant heads for each timestep.

MODFLOW-2005 has three internal nested loops, the stress period loop (outer), time step loop (intermediate), and iteration loop (inner). A predetermined procedure is implemented at the beginning as a routine setup function to read the domain set-

up (i.e., grid resolution, number of layers, and simulation time), model data in the form of boundary conditions, aquifer hydraulic characteristics (i.e., hydraulic conductivity, specific storage, transmissivity), initial head conditions, and selected solution method.

The outer loop is responsible for calculating the resulted heads for each timestep from defined boundary conditions, including

specified heads (i.e., time-variant or head boundary packages), specified flux (i.e., recharge or wells), and head-dependent flux (i.e., drain, evapotranspiration or river recharge). The intermediate loop accounts for the total simulation time, as well as additional output processing, and the inner loop for calculation purposes to approximate the head solution until the maximum number of iterations is achieved. At the end of the iteration loop, specified output control files are created in the form of heads, budget terms, or flow in the domain. The intermediate and outer loops repeat until all timesteps are completed for all stress

periods (Harbaugh, 2005).

FLO-2D model works with variable time steps that are automatically adjusted internally based on stability criteria requirements. Because FLO-2D uses an explicit finite difference method to solve the surface water equations, its time step is





usually much smaller than that defined for the MODFLOW-2005 model, resulting in an increasing number of 2D
computational sweeps to match the MODFLOW-2005 simulation time (FLO-2D, 2018). A time-synchronization scheme was
developed to achieve the coupling, as the MODFLOW-2005 intermediate loop is in charge of transferring the information
between models. For example, the FLO-2D iterative calculations start until reaching MODFLOW-2005 time step one. Then,
the MODFLOW-2005 intermediate loop performs its respective calculations from time step one and is shared in both directions
to continue with the following time step (Nalesso, 2009). The process repeats itself until the simulation time of FLO-2D is
completed. Similarly, MODFLOW-2005 can experience numerous stress periods during the simulation. Fig. 6 depicts the time
step synchronization procedure between both models.

The FLO-2D algorithm calculates the accumulated volume of water that infiltrates from the floodplain before each
MODFLOW-2005 stress time. In this study, the Green-Ampt method (1911) was selected for being the most complete function
to estimate infiltration. With this method, the rainfall intensity predominantly influences the infiltration process as runoff and
is generated when the maximum infiltration capacity is exceeded. Several variables are accounted for in the Green and Ampt
infiltration function, including initial abstraction, hydraulic conductivity, soil porosity, volumetric moisture deficiency (initial
and final soil saturation conditions), soil suction and soil storage depth. The development of the G&A method in FLO-2D is
based on the application of Darcy's Law principle that the infiltration process begins as soon as the surface water moves in a
vertical direction through the permeable medium and can be written as:

$$\frac{\Delta F}{\gamma} - ln\left(1 + \frac{\Delta F}{\gamma + F(t)}\right) = \frac{K_w}{\gamma}\,\Delta t \tag{6}$$

Where:

$\Delta F$ = change in infiltration over the computational time step
$K_w$ = hydraulic conductivity at natural saturation (mm/hr)
$\gamma = (PSIF + Head) * DTHETA$
$PSIF$ = capillary suction (mm)
$Head$ = incremental rainfall for the time step plus flow depth on the grid element (mm)
$DTHETA$ = volumetric soil moisture deficit (dimensionless)
$F(t)$ = total infiltration at time t
$\Delta t$ = computational time step

Fullerton (1983) developed an explicit equation $\Delta F$ by using a power series expansion for infiltration with respect of time to
approximate the logarithmic term in the latter equation:

$$\Delta F = \frac{-[2F(t) - K_w\Delta t] + [(2F(t) - K_w\Delta t)^2 + 8K_w\Delta t(\gamma + F(t))]^2}{2} \tag{7}$$





Conversely, the water exchange can also occur in the opposite direction due to flash flood events, fast recharge, or high-water surface levels in channels due to a sudden rise in the water table. If the groundwater heads calculated in MODFLOW-2005 are higher than the surface depth in FLO-2D, the depth of water from groundwater will be added to the surface depth. The infiltration calculation is switched off at each node as long as the saturation condition persists, meaning that infiltration will not be calculated until the soil absorption capacity is reestablished.


### 4.4 Model configuration and set-up

The FLO-2D hydraulic model requires a grid of square cells to represent the topography of the floodplain domain. The structured grid size of the computational domain defines the hydraulic model resolution. The LIDAR DTM was used as source floodplain topographic information, and an interpolation algorithm was implemented to produce a resampled DTM floodplain

model to be used as input elevation of the hydraulic model. The nearest neighbor interpolation method was selected to resample data from the high-resolution 2 m LiDAR to a 20m resolution ($\approx$ 43,000 cells).

In addition to the topographic features, a detailed representation of the built environment is relevant for urban flood modeling to simulate the flow wave propagation dynamics realistically. All buildings in the domain (7827 features) were imported to

the FLO-2D computational domain. The polygon vectors are represented as Area Reduction Factors (ARF = 1) where the grid element surface area is considered impervious and is removed from potential water interactions.

Bathymetric measures were available for the Little Arch Creek River. A 1D hydraulic model with natural cross-sections was imported into FLO-2D extending from NE 143$^{rd}$ Street to structure G-58 located downstream of the Enchanted Forest Elaine

Gordon Park. Official bathymetry from the Biscayne shore, Keystone Island, and Sans Souci canals was not available for this study due to jurisdiction restrictions. To compensate for the missing geometry, aerial imagery Google Earth was used to measure the canal's width, while a 10-meter bottom elevation was used as constant depth based on the Miami Florida Intracoastal Topography database from the Oleta River.

The infiltration method selected for the case study was the Green-Ampt, and the global soil parameters correspond to the pavement and the porous limestone environment to account for the surface water and groundwater interactions. For simplicity, the Manning roughness coefficient was assumed as 0.40 for green land cover areas and 0.04 for the impervious urbanized environment, canals bed, and Biscayne coast. Rainfall and tides were considered for the hydrologic forcing, setting the precipitation over the whole domain and tide levels in the Biscayne Bay's easternmost cells.

Concerning MODFLOW-2005, a simple model was developed based on two groundwater models, the regional MDC by USGS (Hughes & White, 2016) and the Arch Creek Basin (Sukop et al., 2018). The aquifer is composed of one-layer of about 36 meters, hydraulic conductivity parameters (Kx, Kz, Ss, Sy, and initial head), and four boundary conditions. The CHD package





feature in the easternmost boundary represents the tide conditions of the Biscayne Bay, and the ocean-side water levels from Canal C-8 in the southern boundary edge. In respect to the groundwater heads, the GHB package was used to set the water table levels from gauge station G-852 in the westernmost boundary of the domain.

After the models are set-up, the compatibility process validates the perfect agreement between grid structure, position, and vertical elevations. A perfect match between the surface layers is required for the loosely-coupled model to link the floodplain-aquifer hydrodynamics. If so, FLO-2D will act as the base hydraulic model capable of simulating rainfall and discharge, ocean levels, and groundwater elevations, with the support of MODFLOW-2005, to create a compound inundation model (Fig. 7).

### 4.5 Flood events

Three flood events (2-4 October 2000, 6-8 June 2013, and 25 May 2020) characterized by similar high intensity rainfall, low storm surge levels, and unusually high-water table levels with different response times were selected to compare the surface-subsurface model results (Fig. 8).

On 2-4 October 2000, Tropical Storm Leslie was responsible of one of the most severe events of North Miami in recent history in terms of flooding and property damages, with an accumulated rainfall of 454 mm over 65 hours and an estimated return period of 50 years. The highly permeable limestone and hydraulic conductivity of the Biscayne Aquifer is sensitive and strongly influenced by the direct and rapid response of ground-water levels to local scale rainfall events. As a result, a large area covered by heavy showers in Broward and MDC contributed to the sharp increase in the water table levels prior to the localized precipitation (≈ 20 hours) in the Arch Creek Basin (Franklin et al., 2001).

Similarly on 6-8 June 2013, Tropical Storm Andrea was a short-lived storm that formed in the Gulf of Mexico which produced very heavy precipitation across Broward and MDC (Beven II, 2013), recording a storm total rainfall of 317 mm in the Arch Creek Basin. The intense precipitation over 11 hours promoted the groundwater recharge rates significantly which led to a sudden increase of 1 meter in the water table.

The 25 May 2020 event is categorized as a 25-year storm with a total daily rainfall depth of 263 mm, producing localized rainfall in the North Biscayne Bay watershed, specifically in the Arch Creek Basin. Although the rainfall intensity and peak flow per-se did not represent a major threat to the study site and the storm hydrograph is less severe compared to Tropical Storm Leslie and Andrea, antecedent rainfall and soil moisture conditions exacerbated the magnitude of this event. Low-intensity storms contributed to the consistent recharge of the aquifer since mid-April 2020. It should be noted that May 2020 is also considered the second wettest May on record. As a result, the 25 May 2020 storm resulted in the fast gradual increase



of groundwater table levels from 0.7 to 1.55 meters (NAVD 88) in ≈ 60 hours, leading to a CF event from pluvial and
       groundwater sources.




## 5 Results

**5.1 Compound simulation**

The interaction of groundwater and surface water physical processes are relevant and meaningful to better assess the severity of CF risks in low elevation coastal karsts environments. Fig. 9 illustrates substantial flood magnitude differences when the subsurface hydrology and the infiltration depth are omitted from the 2D flood modelling framework. While the joint impact of rainfall and tide levels per se do not pose significant threats to infrastructure as the surface runoff rapidly infiltrates into the

porous permeable soil (Fig. 9a, 9c, 9e), the shallow water table of the Biscayne Aquifer quickly responds to high-intensity short-duration storms which results in the sudden increase of groundwater levels, leading to extensive urban flooding in parts of the Arch Creek Basin (Fig. 9b, 9d, 9f). The simulation proves reasonable in terms of maximum flood depth and extent due to the similarities in the hydrologic conditions, being Tropical Storm Leslie the most severe of all three storms.

Fig. 10 shows the flood mapping results and the water table timeseries for Tropical Storm Leslie. Although rainfall-runoff is the primary source of flooding in the urbanized Arch Creek Basin, abnormally high groundwater levels triggered groundwater-induced flooding resulting in the amplification of chronic flooding near historic waterways or zones below the County's land elevation flood criteria within North Miami and Unincorporated MDC, with flood depths ≈ 1 meter (Fig. 10a, 10b). The groundwater plots illustrate the effect of tidal and groundwater boundary conditions on the behavior of the simulated water

table, in turn demonstrating the importance of both variables in the modeling set-up and influence in subsurface dynamics, as a cyclic high-low pattern characterize the tide fluctuations of the Biscayne Bay (Fig. 10b – 10e) compared to the defined water heads behavior from well G-852 in the western boundary of the domain (Fig. 10a, 10f).




## 5.2 Identification of flooding hotspots

Despite the absence of post-disaster mapping products, measurements, and limited crowdsourced data in the study area, FEMA's SRL records were used to compare the model results with flood observations. Although the available records do not specify the observed inundation depths, an agreement between the property locations and maximum water levels may offer

sufficient validation to identify properties and neighborhoods at risk where shallow water tables can exacerbate the flooding conditions. The simulated storm events illustrate that most of the properties experienced moderate to high flood depths in predefined locations. For example, the housing infrastructure of Unincorporated MDC are particularly vulnerable to the impacts of surface flooding, even when the water table remains below the surface (Fig. 10b – 10e). Fig. 11 presents a consistent agreement between the reported claims and localized flooding, indicating that the housing infrastructure in these

neighborhoods have experienced SLR and are likely to experience additional flood losses at some point in the future. In terms of residential damage, Tropical Storm Leslie and Tropical Storm Andrea may be considered the costliest events in the Arch Creek Basin as both account for the 60% of the reported claims (25 and 17 respectively) (Table 2).

Sources of uncertainty in the coupled numerical model could be reduced by increasing the model's resolution and incorporating

storm-water infrastructure features. For example, the increase of the water table levels could challenge the ability of the storm drain system to convey water towards the Bay, resulting in prolonged flooding conditions, or anti-flood pump stations may alleviate the impacts of flooding by draining water from the streets and swales back to the ocean. Nevertheless, these records only reflect a small percentage of the damaged infrastructure and cannot be generalized at the Basin scale as the property owners may not meet the criteria to file the claim.


## 5.3 Validation using crowdsourced data from Tropical Storm Andrea

A limited number of real-time and post-flood crowdsourced flooding observations in the Arch Creek Basin were available during Tropical Storm Andrea (Fig. 12). The visual comparison indicates a consistent spatial agreement between the maximum flood depth of the coupled simulation and the interpreted depth of the crowdsourced data (Table 3).

Fig. 12a associates high flow depths (> 0.5 meters) with several properties that have experienced regular chronic flooding conditions, while the crowdsourced photograph displays an estimated inundation depth of 0.20 meters. Despite the model's overestimation, this comparison can be seen as an effective form of validation considering the changes in land use associated with the Arch Creek flow (Fig. 1c) and low topographic elevation (Fig. 3b).

Regarding Fig. 12b, the US Post Office exhibits chronic flooding in the parking lot. We observe a reasonable level of accuracy

in terms of flood depth validation results derived from the coupled model. Fig. 12c displays stagnant flood water accumulated





after the event in a portion of the NE 14 Ave. The results suggest that the rise in the water table influence the inundation extents, water levels and flood timeline.

Although the limitations on the amount of collected crowdsourced data in the study area, a larger georeferenced dataset including the date and time could improve the reliability of VGI data to validate hydrodynamic models. Similarly, a higher
spatial resolution could reduce the level of uncertainty and biases from the modelling results.



## 6 Discussion

### 6.1 Flood risk and vulnerability

Floods resulting from extreme weather and climate events represent a major threat to low-lying neighborhoods and housing infrastructure in the Arch Creek Basin. Historically, frontal systems and summer cloudbursts are responsible for most of the significant pluvial flooding events in the study area compared to strong tropical systems, with Hurricane Irene (1999), Katrina (2005), Irma (2017), and No-Name storms as the only exceptions (Miami-Dade, 2015).

Most of the population of MDC lives in high-risk areas, only 1.2 meters (4 feet) above sea-level. In regard to the Arch Creek
Basin, three-quarters of the urban landscape (67%) are located in a 100-year flood-prone area, and over 80% of the housing stock was built prior the development of the 1973 Flood Insurance Rate Map (Miami-Dade, 2016). For instance, properties in the Arch Creek Estates and localized areas East of US-1 such as the Key Stone Islands and Sans Souci Estates experienced repetitive flood losses since these settlements were built in the former riverbed of the Arch Creek Rivers or in land reclamation areas. The capacity of these communities to respond to hydrometeorological phenomena is limited or non-existent, resulting
in repetitive negative impacts on livelihoods and residential property, expanding the socio-economic gap and inequality of MDC communities (Keenan et al., 2018).

### 6.2 Groundwater level fluctuations

The results of this investigation determined that groundwater tables rise rapidly with rainfall events leading to surface flooding in the Arch Creek Basin. Similar results were obtained by Sukop et al. (2018) who found that precipitation as the main trigger
for groundwater-induced flooding, with tidal fluctuations and sea level rise increasing the shallow water table, contributing to the reduction of the storm drain capacity. The present study also determined that antecedent rainfall events were important in the height of the water table at the start of the rainfall events investigated.

Seasonal water table fluctuations are expected throughout the year, presenting a higher level frequency during the winter and
spring seasons due to climate variability and hydrological forcing (Gurdak et al., 2009; Taylor & Alley, 2001). Nevertheless, as we observed with Tropical Storm Leslie and Tropical Storm Andrea, the potential rise of groundwater levels to the surface during dry season cannot be ruled out since the hydraulically non-restrictive nature of the carbonate strata in MDC allows for rapid infiltration and high recharge rates during heavy precipitation events. The hydrologic forcing input and modeling results suggest that the joint occurrence of a high-intensity short-duration precipitation (> 50 mm peak, 250 mm total) with already
high groundwater levels (> 1 meter) result in a CF event. Further research on linking multivariate statistical analysis with coupled hydrodynamic modeling frameworks may prove beneficial to identify thresholds that trigger CF conditions (Couasnon et al., 2018; Jane et al., 2020; Moftakhari et al., 2019; Saksena et al., 2019; Sebastian et al., 2017).





## 6.3 Tides and sea level rise

MDC is already experiencing the cascading effects of climate change with a record of 39 high-tide flooding events in 2019
(Wdowinski, 2019), costliest and most active hurricane seasons in records (2017 and 2020) (NOAA, 2021), steady increase in
higher water table anomalies since 2010 (SFWMD, 2021), continuous saltwater intrusion (Guha & Panday, 2012; Obeysekera
et al., 2011), and more frequent groundwater flooding events (Compact, 2020; Sukop et al., 2018). Current SLR projections
are expected to amplify future flood hazards in MDC including the variability in hydrological processes and extreme events,
as well as the frequency and duration of nuisance flooding and shallow water tables (Obeysekera & Salas, 2016; Sweet et al.,
520  2016).

Although this investigation determined that rainfall and sea levels alone did not produce significant flooding, the modeling
efforts did not include storm surge flooding that can often accompany large hurricanes (Zhang et al., 2013). Nonetheless SLR
projections and induced storm surge flooding conditions are beyond the scope of this study, future work on assessing the
impact of high tide and storm surge induced flooding are fundamental to assess CF events and future flood risk scenarios
(Obeysekera et al., 2019).

## 6.4 Wastewater and pollutants

Understanding the potential for groundwater tables to rise above the ground surface is important as Smith et al. (2021)
determined that rising groundwater tables can carry contaminants from below ground septic systems to surface waters.
Wastewater in septic systems often contain fecal coliforms, nitrate, phosphate as well as a number of pharmaceutical
compounds such as antibiotics, analgesics and synthetic hormones (Yang et al., 2016). Rising groundwater tables not only
present a concern to property damage as documented in this investigation, but also raise concerns for human exposure to
wastewater pollutants. Furthermore, as floodwaters run-off, they can transport pollutants to adjacent surface water bodies such
as Biscayne Bay. Wastewater contaminants have been found to persist in south Florida coastal waters (Singh et al., 2010).
Future flood management efforts should consider flood water treatment to alleviate polluting adjacent surface waters.





## 7 Conclusions

Compound flooding hazards are increasing in coastal cities due to multiple factors related to climate change. The Arch Creek Basin in North Miami, which served as a vital flow corridor that connected the Everglades to the Biscayne Bay, is an

appropriate location to study CF conditions. Results corroborate that groundwater-induced flooding is localized; thus, becoming an underlying condition that must be considered in low elevation coastal karst environments where the water table dynamics are subject to swift fluctuations caused by rainfall events.

A knowledge gap regarding a consolidated groundwater modelling framework was identified and addressed by proposing a

loosely-coupled flood model that integrates surface hydrology and groundwater. The ability to produce more comprehensive flood hazard mapping from couple surface and subsurface water interactions is scientifically relevant to professionals in hydroinformatics since it improves the replicability of flood dynamics, setting the path to improve the understanding, prediction, and response time of groundwater levels as a potential trigger to compound flooding phenomena that can exacerbate floodwater depth and areal extent. This work opens new horizons on the development of CF models from a holistic perspective.


The quality and accuracy of flood hazard mapping in urban areas are strictly related to the model spatial resolution considering that the vertical datum and built-up environment influence flow propagation dynamics. A 20-meters grid resolution was selected to balance the computational demands with a certain level of precision without compromising the quality of the simulation. However, the investigation of higher and coarser resolutions in CF studies might yield insights into the estimation

of inundated areas and time performance at different scales.

Considering Miami's hydrogeomorphology is one of the most complex globally, the compounding effects of flood drivers may respond differently in diverse geographic settings. Therefore, further research should consider the proposed modeling framework to assess the CF risk in different geographical regions prone to multiple flood drivers, specifically in areas that

have access to post-event flooding maps in the form of remote sensing products or VGI data for calibration and validation purposes.

The ability to simulate rising groundwater levels and sea level rise will be of great interest to Miami-Dade authorities on the impact of flooded septic systems from an ecological and public health perspective, providing a clearer view on the spread of

septic tank effluent and contamination hotspots.

The contributions of this research are substantial and go beyond the numerical simulation scope, as it supports numerous fields and real applications including flood management, urban planning and design, flood mapping and zoning, disaster risk reduction, flood insurance policies and policy making. Ultimately, this research is a small piece of multidisciplinary work that



analyzes the ripple effects of flooding in a wide range of fields (such as socio-economic costs, urban and ecological degradation, and health) and can set the basis for prevention, protection, accommodation, and even retreat/relocation policies.

*Author contributions.*

FP and JO jointly conceptualized the research experiment, from the design of the procedure to the presentation of results. FP
gathered and processed the case study data, developed the coupled model framework, calibrated, and validated the simulation results, wrote initial version of manuscript, and produced all figures and tables. NGR provided the technical expertise to achieve the coupling between FLO-2D and MODFLOW-2005. FN, JO, and AM provided guidance and supervised the work of FP. RP and FC shared ideas to improve the results and discussion sections. FN, AM, JO, RP, FC, and TC contributed to the paper revisions.


*Competing interests.* The authors declare that they have no conflict of interest.

*Acknowledgements.* We gratefully acknowledge Marcia Steelman from MDC for the kind support throughout this research, including the provision of detailed background of the study area, documentation, historic imagery, shapefiles, and
crowdsourced data. We thank Angela Montoya from MDC for her helpful assistance on understanding MDC's regional groundwater model using MODFLOW, and Ruben Arteaga from SFWMD for sharing flood protection and planning drainage reports. We would also like to thank our colleagues Michael C. Sukop and Martina Rogers from FIU for their valuable tutoring and recommendations during the development phase of the Arch Creek MODFLOW model.

*Financial support.* This work was supported by the University for Foreigners of Perugia— ISPRA INFO/RAC2020 Research Grant No. COAN AC.11.04.01 (Research grant "Research and implementation of GIS and hydrologic-hydraulic models for large scale water and flood risk management to support the Disaster Risk Reduction program"). In addition, this material is based upon work supported by the National Science Foundation under Grant No. HRD-1547798. This NSF Grant was awarded to Florida International University as part of the Centers for Research Excellence in Science and Technology (CREST)
Program. This is contribution number 1024 from the Southeast Environmental Research Center in the Institute of Environment at Florida International University. This work was also funded by Florida International University Sea Level Solution Center Grant No. 800008174, and the Dissertation Year Fellowship from the FIU University Graduate School.



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

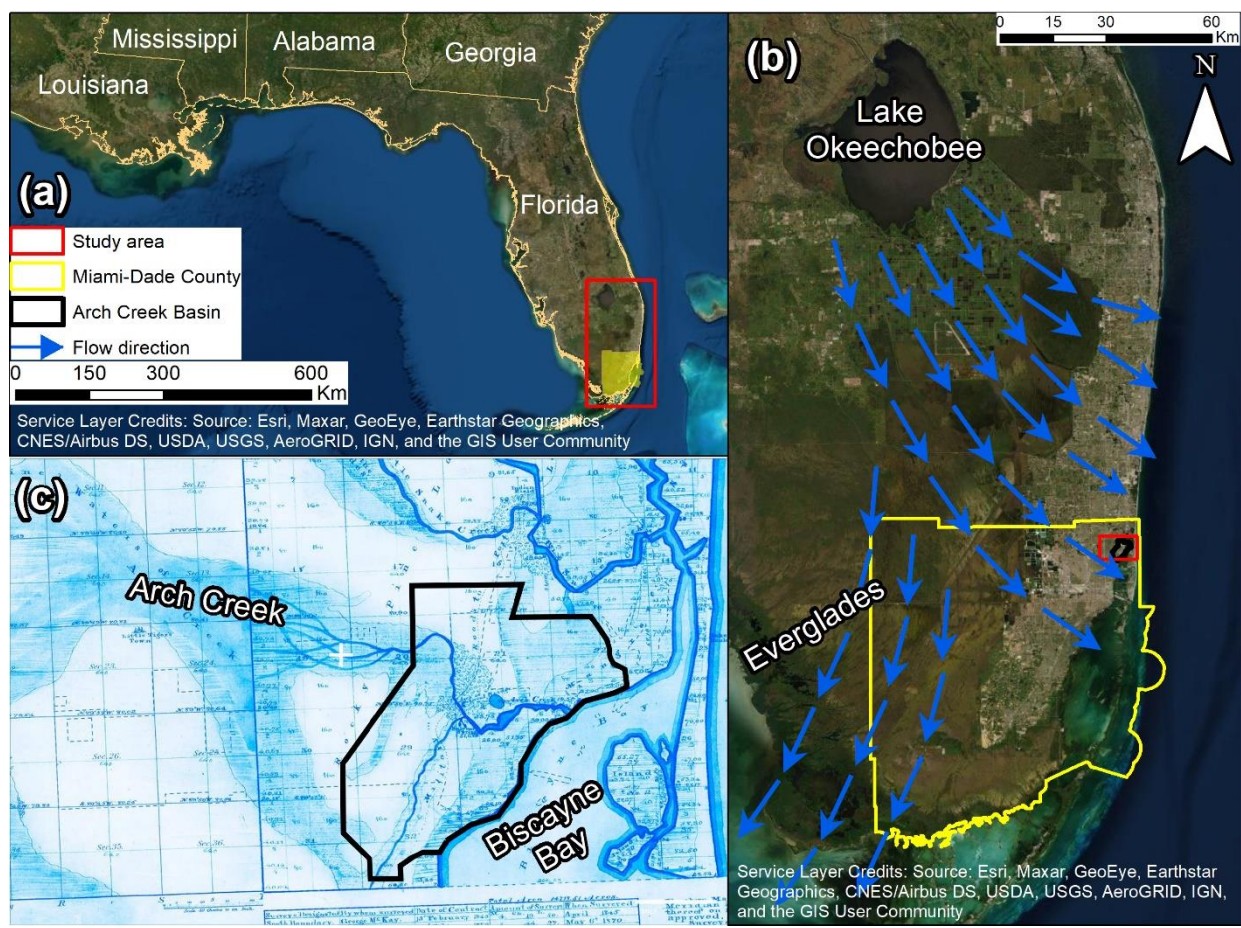

**Figure 1.** Location map of the study area. (a) MDC located in Southeast Florida, USA (b) current Everglades water flow from Lake
Okeechobee towards the Atlantic Coast and Gulf of Mexico, and (c) land survey from 1870 that illustrates the natural flow direction of the
Arch Creek to discharge into the Biscayne Bay prior urbanization (Miami Herald, 2019).





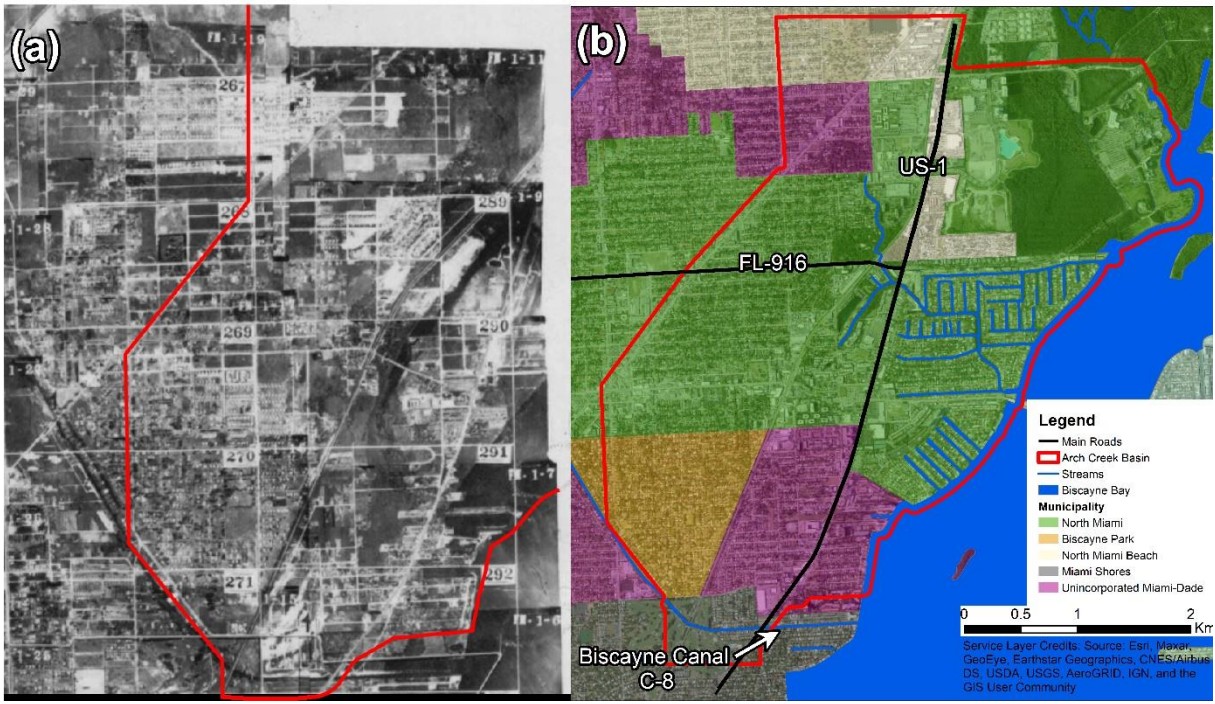

**Figure 2.** Aerial photography of historical (1948) and current urbanized environment in the study area. (a) Major civil and drainage works contributed to the rapid urbanization of the Arch Creek Basin; (b) Municipality map, including North Miami, Biscayne Park, North Miami Beach, Miami Shores and Unincorporated Miami-Dade (U.S. Department of Agriculture, 1948).

**Table 1.** Population and land elevations of Arch Creek Basin jurisdictions. Population totals account for the whole jurisdiction area (U.S. Census Bureau, 2020)

| Jurisdiction | Population* | Area (km²) | Area ACB (km²) | Percentage of land elevation (meters) | | | | |
|---|---|---|---|---|---|---|---|---|
| | | | | < 0 | 0 - 1 | 1 - 2 | 2 - 5 | > 5 |
| North Miami | 62489 | 26.09 | 11.00 | 7.88 | 18.64 | 39.67 | 31.27 | 2.54 |
| Biscayne Park | 3124 | 1.64 | 1.44 | 0.00 | 1.48 | 77.20 | 21.32 | 0.00 |
| North Miami Beach | 42971 | 13.79 | 1.43 | 0.01 | 11.53 | 20.05 | 68.41 | 0.00 |
| Miami Shores | 10459 | 9.80 | 0.54 | 4.82 | 19.68 | 38.91 | 36.56 | 0.03 |
| Unincorporated MDC | N/A | 25467 | 2.54 | 3.65 | 14.60 | 47.08 | 34.67 | 0.00 |





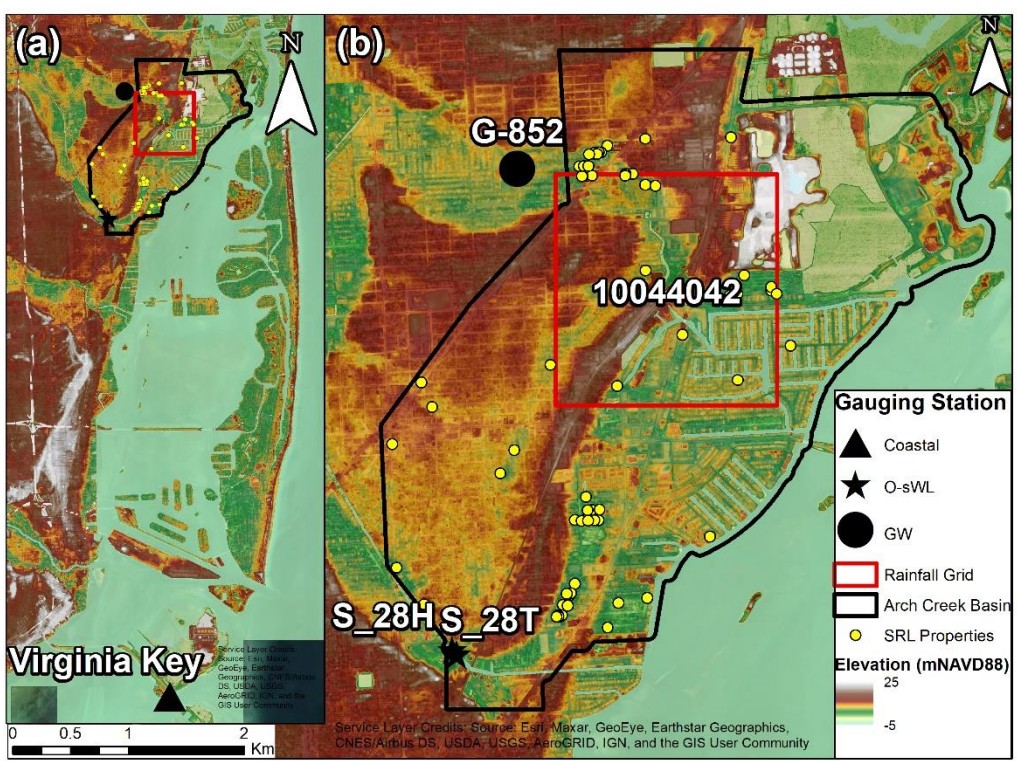

**Figure 3.** Geographical location of selected data in the study site. (a-b) Topographic map showing the location of the Arch Creek Basin (black polygon), and the distribution of closest gauging stations to the study site (black), rainfall grid (red square), and FEMA's SRL claims
(yellow).

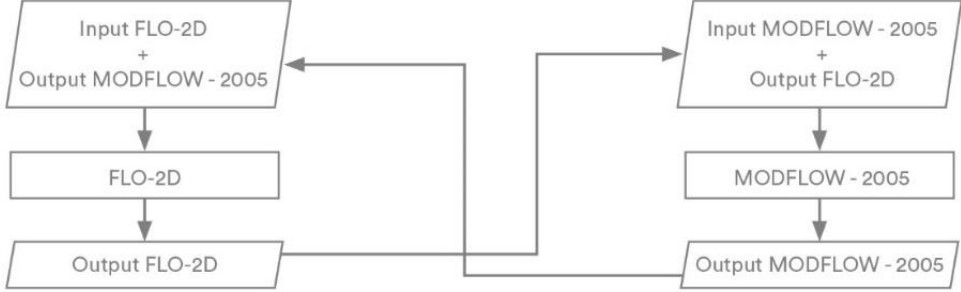

**Figure 4.** Flowchart representing the loosely-coupled joining technique between FLO-2D and MODFLOW-2005.



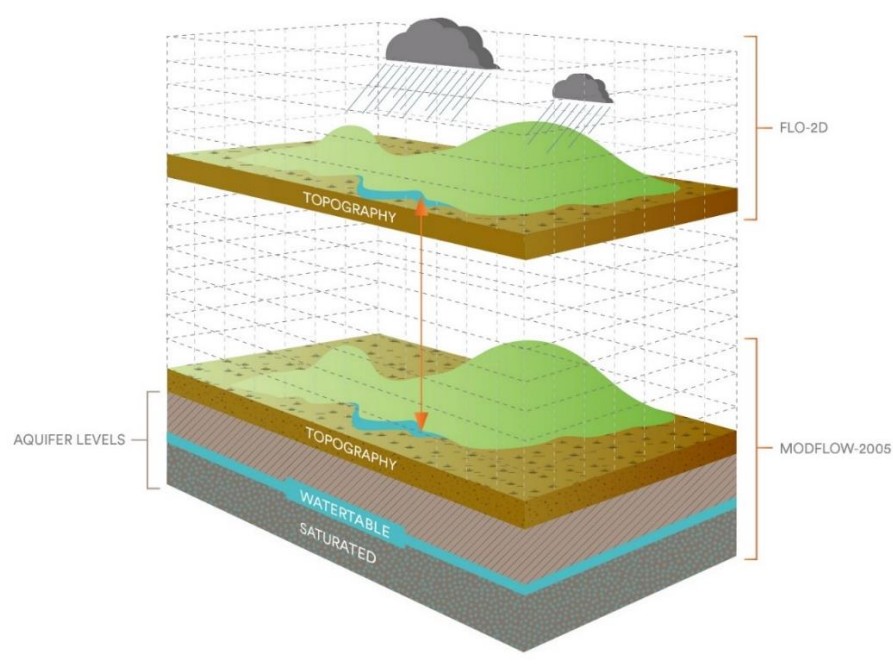


**Figure 5.** Spatial compatibility between FLO-2D and MODFLOW-2005

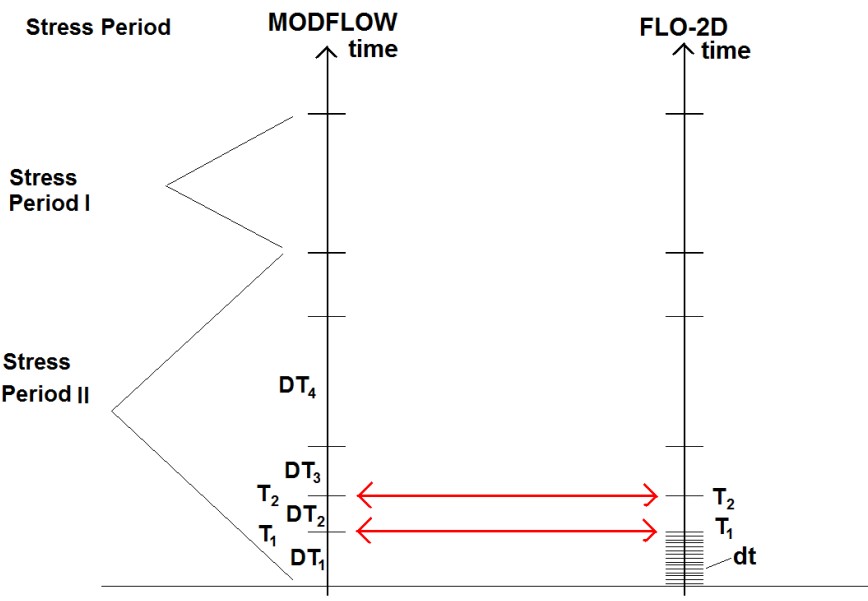


**Figure 6.** Time-step synchronization of FLO-2D and MODFLOW-2005



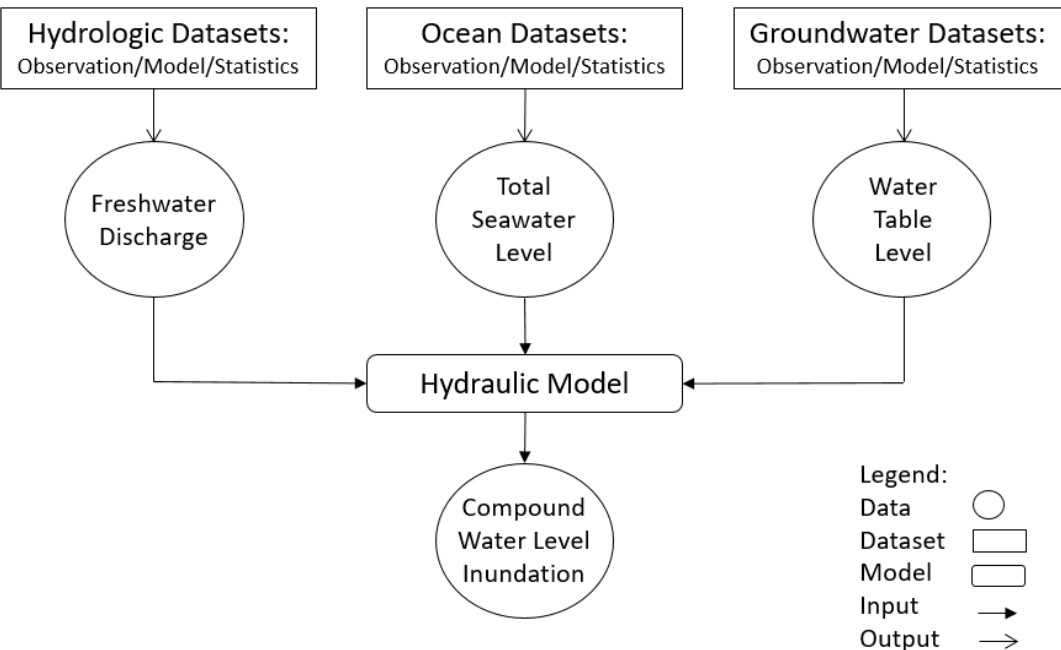

**Figure 7.** Flowchart representing the CF simulation using FLO-2D as the based hydraulic model to connect hydrologic, ocean and groundwater datasets, the latter with the support of MODFLOW-2005. Adapted from Santiago-Collazo et al. (2019)




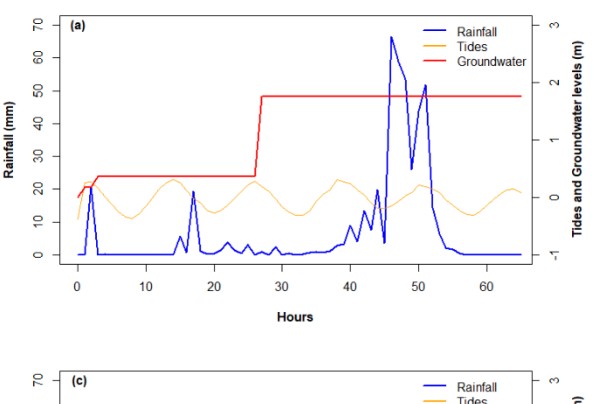

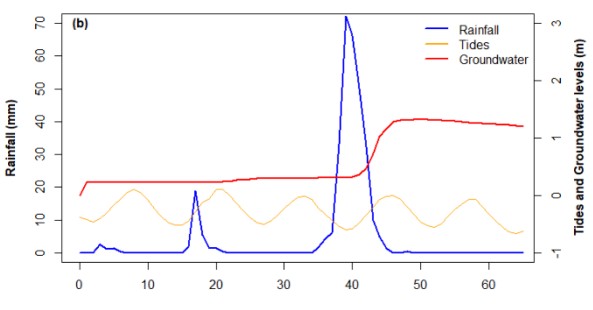

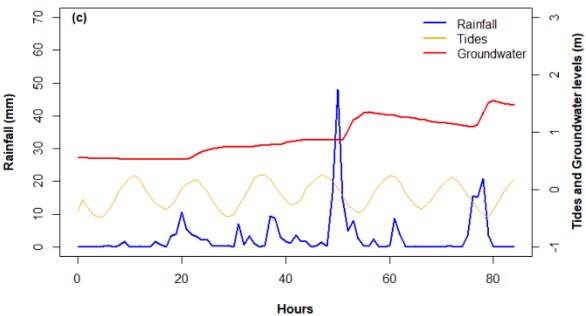

**Figure 8.** Time series of rainfall, tides, and groundwater levels for: (a) Tropical Storm Leslie; (b) Tropical Storm Andrea; (c) 25 May 2020 storm. The simulation time was determined based on the rainfall duration and groundwater fluctuations to properly characterized each event, being 64-hours for both Tropical Storms and 84-hours for the May 2020 event.

**Figure 9.** Spatial distribution of maximum inundation depths for rainfall and tides (left) and the compound flooding interaction of rainfall, tides, and water table (right) for Tropical Storm Leslie (a-b), Tropical Storm Andrea (c-d), and 25 May 2020 event (e-f).



**Figure 10.** Distribution of maximum water surface elevations and groundwater table profiles in six sample locations across the Arch Creek Basin for Tropical Storm Leslie.

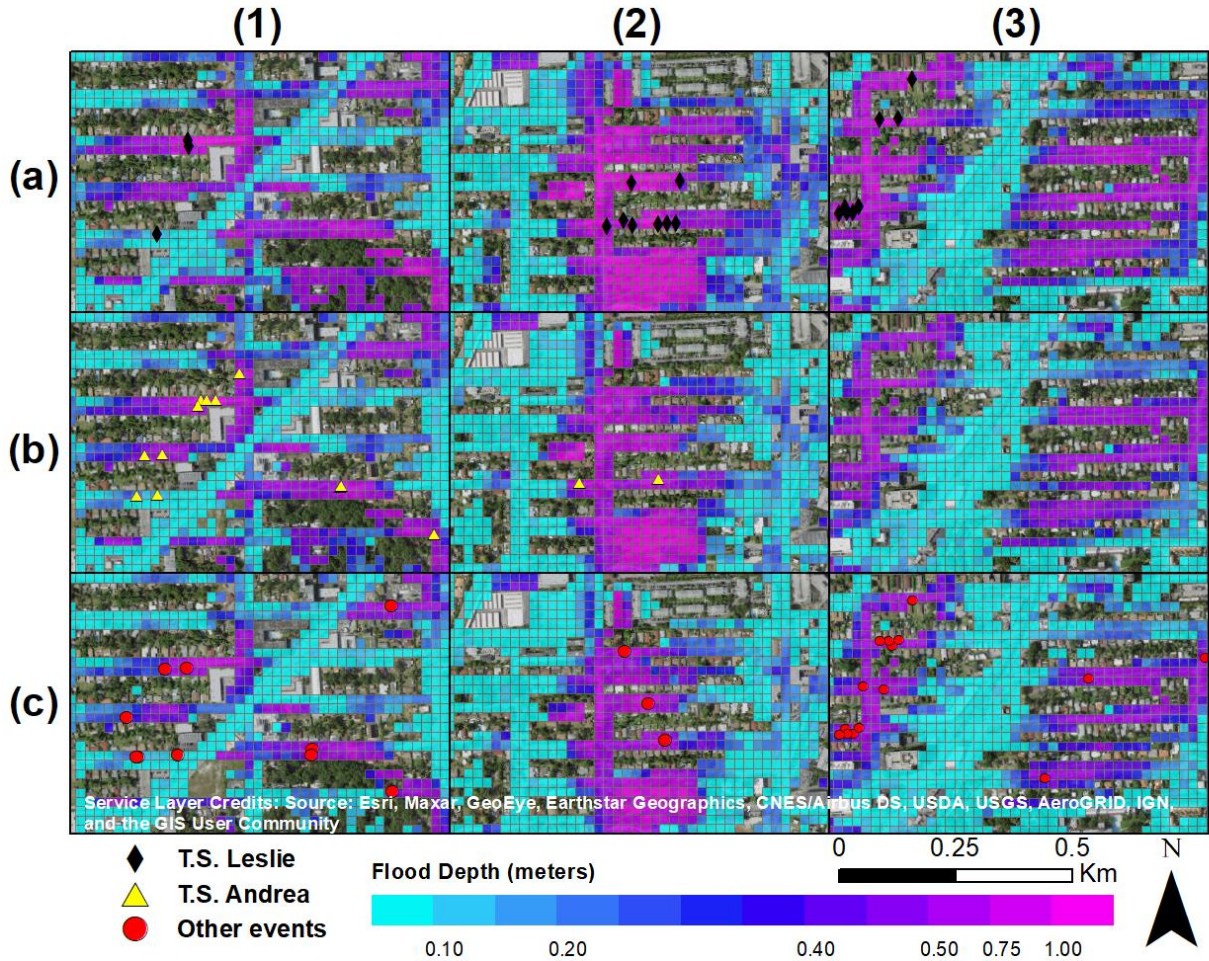

**Figure 11.** Distribution of maximum water surface elevations in three sample locations (Fig. 9) for Tropical Storm Leslie (a), Tropical Storm Andrea (b), and 25 May 2020 event (c) against FEMA's SRL database (yellow). A high rate of agreement between FEMA's claims and high flood depths is achieved by the compound flood simulations.




**Table 2.** Quantitative analysis of simulated flood depths in respect to FEMA's SRL database by events.


| Flood depth (mts) | T.S. Leslie | T.S. Andrea | Other Events |
|:---:|:---:|:---:|:---:|
| 0 - 0.1 | 2 | 0 | 3 |
| 0.1 - 0.2 | 1 | 1 | 5 |
| 0.2 - 0.3 | 0 | 1 | 3 |
| 0.3 - 0.4 | 1 | 1 | 5 |
| 0.4 - 0.5 | 0 | 2 | 5 |
| 0.5 - 0.75 | 4 | 5 | 10 |
| 0.75 - 1.0 | 13 | 7 | 2 |
| 1.0 - 2.0 | 4 | 0 | 0 |
| Total | 25 | 17 | 33 |

**Table 3.** Comparison between simulated maximum water flood depths and VGI imagery obtained during and after Tropical Storm Andrea.

| No. | Latitude | Longitude | Image category | Interpreted depth (m) | Max simulated depth (m) | Difference (m) |
|:---:|:---:|:---:|:---:|:---:|:---:|:---:|
| 1 | -80.165579 | 25.910225 | During storm | 0.20 | 0.67 | -0.47 |
| 2 | -80.157365 | 25.908227 | During storm | 0.55 | 0.54 | 0.01 |
| 3 | -80.170807 | 25.900715 | After storm | 0.25 | 0.23 | 0.02 |


**Figure 12.** Maximum surface water depths of Tropical Storm Andrea in the Northwestern portion of the Arch Creek Basin (top right). Three selected subdomains (left) with available crowdsourced observations (white) are compared against FEMA's claims (yellow) and the simulated groundwater levels, resulting in rainfall-induced flooding as the water table remained below the terrain elevation (brown).
