# Peer review of "Compound flood modelling framework for rainfall-groundwater interactions"

_Natural Hazards and Earth System Sciences, 2021_

## Author Comment (AC1)

**Response to Anonymous Referee 1**

**1. This manuscript introduces a coupled model to jointly simulate groundwater levels and surface flows to assess compound flooding in South Florida. The two models that are loosely coupled are FLO-2D and MODFLOW-2005.**

**The modelling framework is applied for three past storm events to assess flood extent and depth and the role of the different flooding drivers. Validation opportunities are limited due to missing flood inundation information, but ancillary data is collected and used to assess the model performance.**

**Overall, I think the manuscript is very interesting, timely, and generally well written. I have some general and several specific comments listed below which I think should be taken into consideration to improve the analysis/presentation.**

**Authors' comment:** We thank Reviewer #1 for the positive feedback.

**Action taken**: The authors would like to thank the reviewer for providing thoughtful comments. Please find our responses to each comment below including how we plan (following the NHESS review process) to adjust the manuscript.

**General comments:**

**2. I think the authors need to be clearer in how they define compound flooding for the purpose of their analysis, i.e. is a rainfall event that coincides with a high water table considered a compound event or do tides also have to contribute to escalate the flood depth/area to make it a compound event?**

**From the introduction it appears that the key focus is on the combination of all three flooding drivers, rainfall, ground water, and coastal water levels, but the latter barely play a role in the events that were analyzed. I am not saying that the combination of rainfall and high water table could not be considered a compound event, but there was a bit of a mismatch of what I read in the introduction/methods and what was shown in the results.**

**I think this can be fixed by changing the narrative and does not require additional analysis.**

**Authors' comment:** We thank Reviewer #1 for the suggestion. This paper aims to present an integrated modeling framework capable of simulating surface-subsurface water interactions. For the purpose of this analysis, compound flooding is defined as the interaction of overland flow and groundwater emergence to the surface. For this reason, the manuscript title has been modified to " Compound flood modelling framework for surface-subsurface water interactions" and the narrative was modified to be clearer and improve the readability of the manuscript.

**Action taken**: The narrative has been clarified in the following sections:
Abstract (lines 19-31)

[revised manuscript text omitted]

**3. I am not familiar with the exact models that were used for the analysis and hence I sometimes had a hard time understanding in the Methods section which parts of the modelling system were actually developed by the authors and which parts were already implemented. Much of it read like material I would expect in the technical manuals of the models. I would keep such information at a minimum and only provide the information**

**necessary to understand the new aspects of the framework going beyond what was already there, and rather refer to the technical documentation for details of how these models work.**

**Authors' comment:** We agree with Reviewer #1 that the methodology section contained redundant information about both modelling frameworks and the coupling methodology is not properly explain and requires editing

**Action taken**: The methodology sections (4.1 to 4.4) has been fully restructured as requested (lines 208 - 369)

**4. The figure captions are all very short and often don't contain enough information to fully understand what is actually shown; this is particularly important since many of the figures are stitched together and contain a lot of information. See also specific comments below regarding figures/legends.**

**Authors' comment:** We agree that the figure captions are not self-explanatory and some figures require editing

**Action taken**: Most figures and captions have been improved as requested

**Specific comments:**

**5. Line 19- "Physics-based" might be better**

**Authors' comment:** We agree

**Action taken**: The sentence has been restructured as requested

**6. Line 22- "returns"**

**Authors' comment:** We agree

**Action taken**: The sentence has been restructured as requested

**7. Line 24- "that as" doesn't really work here I think**

**Authors' comment:** We agree

**Action taken**: The text has been deleted as the narrative has changed

**8. Line 25 "result"**

**Authors' comment:** We agree

**Action taken**: The text has been deleted as the narrative has changed

**9. Line 27 "damage of"**

**Authors' comment:** We agree

**Action taken**: The sentence has been restructured as requested

**10. Line 44 remove "field"?**

**Authors' comment:** We agree

**Action taken**: The sentence has been restructured as requested

**11. Line 50 there is a recent paper by Gori et al. (https://doi.org/10.1029/2019WR026788) which could be added to the list**

**Authors' comment:** We agree

**Action taken**: The citation has been added to the list as requested

**12. Line 73 "in that study"**

**Authors' comment:** We agree

**Action taken**: The sentence has been restructured as requested

**13. Line 165-171 Do I understand correctly that total still water levels (tide + surge) are considered? I was confused when I read later in the discussion/conclusion that storm surges were not included in the analysis. If still water levels were used but from "calm" periods where surge component was minimal that should be made clear.**

**Authors' comment:** We thank Reviewer #1 for the valuable suggestion. Tide levels were considered as part of the coastal boundary conditions. The three selected events are characterized by 'regular' tide level conditions that have little impact on the coast. Therefore, the surge component is minimal and does not exacerbate flood risk conditions in the study area.

**Action taken**: This concept has been clarified as requested in the following sections:
Introduction: Lines 100-104

For the purpose of this analysis, three events characterized by short-lived heavy precipitation, regular tide levels and unusually high-water tables were selected to demonstrate the importance of simulating surface-subsurface water interactions in urbanized karst coasts, as high groundwater heads may exacerbate flooding conditions. In

the context of this paper, compound flooding is defined as the interaction of overland flow and groundwater emergence, while surge levels are normal and have a minimal influence in the inundation beyond the coast.

Methodology: Lines 324-326

Rainfall and tides were considered for the hydrologic forcing, setting the precipitation over the grid system and tide levels in the easternmost cells to represent the Biscayne Bay's coastal conditions. Both time series are structured on a one-hour basis and are presented in the following section.

Methodology: Lines 371-372

Three flood events characterized by similar high intensity rainfall, tide levels, and unusually high-water table levels with different response times were selected to compare the surface-subsurface model results (Fig. 8).

Results: Lines 387-388

Tide levels per se do not pose significant threats to infrastructure as the coastal waters remain within the channels.

**14. Line 199 would drainage systems be considered as sinks or explicitly modelled?**

**Authors' comment:** We thank Reviewer #1 for bringing this up.

The FLO-2D components are explicitly modeled, the interaction between the grid cell (surface model) and the components are physically represented and modeled.

For example, the storm drain is simulated as a fully integrated system on a computational timestep basis. The FLO-2D model moves around blocks of water on a discretized grid system. Grid elements assigned as inlets/outfalls connect the surface layer with the closed conduit storm drain system. A comparison of the grid element water surface elevation with the pressure head from the closed conduit system node in a given cell determines the direction of the flow exchanged between the two systems. Discharge from FLO-2D surface layer to FLO-2D storm drain layer is based on the inlet geometry and water surface depth, return flow is only allowed when Pressure head is greater than water surface elevation. The system is a physical system that represents the real interaction between surface and storm drain layers.

It should be noted that the storm drain component (mentioned in the methodology section) serves to only highlight the model versatility to simulate flooding conditions in urban environments and was never considered in this study.

Further research should incorporate additional urban features such as the storm drain system and pumps to improve the model's flow propagation dynamics.

**Action taken**: Section 4.1 was shorted to comply with comment 1.3 (lines 208-216). In addition, we acknowledged the limitations of the study and underlined potential sources of uncertainties, such as

the importance of increasing the model resolution and include additional urban features (i.e., storm drain system and pumps) to improve the model's flow propagation dynamics (lines 327-328, 415-421).

Lines 208-216
FLO-2D is a physically-based volume conservation model that combines hydrology and hydraulics to simulate the propagation of water dynamics in urban, riverine, and coastal environments for flood hazard mapping, floodplain delineation, flood vulnerability assessments and mitigation planning (O'Brien et al., 1993). The flood routing model applies the dynamic wave approximation to the momentum equation to calculate the average flow velocity across the square grid system one direction at a time in eight potential flow directions over the floodplain. Hydrological processes are represented as rainfall data over the computational domain or as input hydrographs that can be specified in the channel, floodplain, or along the coasts. Various attributes (elevations, roughness coefficient), components (channel, infiltration, storm drain) and features (streets, hydraulic structures) can be incorporated into the FLO-2D model to produce more refined simulations (O'Brien, 2011). Details are described elsewhere (Annis and Nardi, 2019; Grimaldi et al., 2013; Peña et al., 2021; Peña and Nardi, 2018).

Lines 327-328
The inclusion of the storm drain system, French drains, surface water control structures and pump stations in the modelling framework is beyond the scope of this study.

Lines 415-421
Sources of uncertainty in the coupled numerical model could be reduced by increasing the model's resolution and incorporating storm-water infrastructure features (i.e., French drains). For example, the increase of the water table levels could challenge the ability of the storm drain system to convey water towards the Bay, resulting in prolonged flooding conditions, or anti-flood pump stations may alleviate the impacts of flooding by draining water from the streets and swales back to the ocean. Nevertheless, the repetitive loss records only reflect a small percentage of the damaged infrastructure and cannot be generalized at the Basin scale as the property owners may not meet the criteria to file the claim. Therefore, the presented modelling results fall more on the conservative side and might overestimate the real flooding conditions.

**15. Line 234 I would change to "solvers for matrix equations" to avoid repetition**

**Authors' comment:** We agree

**Action taken**: The sentence has been restructured as requested

16. Line 329 "G&A" has not been defined; this is also an example where I wasn't sure if the authors had added new options/functionality or just selected one of different existing options already available in the model code

**Authors' comment:** We agree

**Action taken**: The term has been defined as "Green & Ampt method" and is now consistent throughout the manuscript. This observation has been previously addressed in comment 1.3

**17. Line 378 should it be "canal bed"?**

**Authors' comment:** We agree

**Action taken**: The sentence has been restructured as requested

**18. Line 382 "what is the CHD package feature"?**

**Authors' comment:** We agree

**Action taken**: The term has been defined as requested (Line 356-357).

**19. Line 397 "responsible for"**

**Authors' comment:** We agree

**Action taken**: The sentence has been restructured as requested

**20. Line 393-416 This is where I started wondering what the actual role of the coastal water level**
will be in the analysis (after a lot was said about it before) since it is not mentioned at all, other than that the events had low storm surge levels.

**Was there a particular reason to not select any events where there was at least some storm surge to actually see the effect?**

**Authors' comment:** We thank Reviewer #1 for bringing this up. As mentioned previously (comment 1.13), the selected events have a minimal influence in the flooding conditions across the domain, as the coastal boundary only increase the water levels of the Biscayne Bay and waterways.

The combination of heavy precipitation and unusually high water table levels with high storm surge was not found in the timeseries records.

Although some exceptions presented heavy precipitation and high storm surge with above regular water table levels, these events did not experience groundwater-induced flooding. Therefore, the selection of those will not be of interest for this manuscript as we are studying the influence of surface-subsurface water interactions.

**Action taken**: This observation has been previously addressed on comments 1.2 and 1.13

**21. Line 427 "karts environments"**

**Authors' comment:** We agree

**Action taken**: The sentence has been restructured as requested

**22. Line 435 I was confused here since it says Fig. 10 shows results for Leslie, but it also has results for other events.**

**Authors' comment:** We agree

**Action taken**: The results section, Figures and associates statements have been improved.

**23. Line 437 How is "chronic flooding" defined here? It's a term often used when analyzing high tide flooding but that is different, I think to what the authors refer to here. Please clarify.**

**Authors' comment:** We thank Reviewer #1 for pointing this out.

**Action taken**: Concepts related to "chronic flooding" have been removed from the manuscript to avoid potential misunderstandings.

**24. Line 441 should be "characterizes" I think**

**Authors' comment:** We agree

**Action taken**: The sentence has been restructured as requested

**25. Line 453 In Fig. 10b the water table actually goes above the terrain for all events at some point, should the reference be to "(Fig. 10c-f)"?**

**Authors' comment:** We agree

**Action taken**: The sentence has been restructured as requested

**26. Line 453 "consistent agreement" sounds a bit strange, maybe reword**

**Authors' comment:** We agree

**Action taken**: The term "consistent agreement" has been changed to "reasonable results"

**27. Line 455 do you mean SRL? I don't understand how any of the results presented here would show the effect of SLR (assuming that it stands for sea-level rise).**

**Authors' comment:** We thank Reviewer #1 for pointing out this concern. We apologize for the confusion that was due to similarities with the abbreviation of sea-level rise (SLR).

**Action taken**: To avoid confusion, we define "SRL" as "Severe Repetitive Loss" throughout the manuscript.

**28. Line 457 "account for 60%"**

**Authors' comment:** We agree

**Action taken**: The sentence has been restructured as requested

**29. Line 478 I think a better way to start the sentence is to use "Despite…" or something similar**

**Authors' comment:** We agree

**Action taken**: The sentence has been restructured as requested

**30. Line 512 the paper by Serafin et al. (https://doi.org/10.5194/nhess-19-1415-2019, 2019) could be added to the list**

**Authors' comment:** We agree

**Action taken**: The citation has been added to the list as requested

**31. Line 515 "on record"**

**Authors' comment:** We agree

**Action taken**: The sentence has been restructured as requested

**32. Line 517 here the authors use again "SLR" and I think this time it stands for sea-level rise, but it has not been defined anywhere.**

**Authors' comment:** We agree

**Action taken**: This observation has been previously addressed on comments 1.27

**33. Line 522 see comment above about using still water levels vs predicted tides as boundary conditions and the choice of picking three events where surge component was small**

**Authors' comment:** We thank Reviewer #1 for pointing this out.

**Action taken**: This observation has been previously addressed on comments 1.2, 1.13 and 1.20

**34. Line 546 "coupled"**

**Authors' comment:** We agree

**Action taken**: The sentence has been restructured as requested

**35. Figure 6: maybe consider switching "Stress Period I" and "Stress Period II" text as you start at the bottom with DT1 and T1 but they are linked to period II.**

**Authors' comment:** The figure required additional editing. We apologize for the confusion due to mistakes in the text.

**Action taken**: Figure 6 (now Figure 5) has been improved as requested

[Figure]

**Figure 1.** Time-step synchronization of FLO-2D and MODFLOW-2005.

**36. Figure 7: in the top what does "statistics" refer to? I didn't see anything about that in the text. In the caption it should be "base hydraulic model"**

**Authors' comment:** The figure required additional editing because the current description can lead to confusion.

**Action taken**: The terms "observations / model / statistics" were removed, and the figure has been improved to better communicate the coupling framework between FLO-2D and MODFLOW-2005 to produce a compound inundation scenario.

[Figure]

**Figure 2.** Flowchart representing the CF simulation using FLO-2D as the base hydraulic model. The hydrologic, ocean, and groundwater datasets were obtained through observations. The surface hydrology was incorporated as rainfall and coastal boundary conditions in FLO-2D. The groundwater heads were calculated in MODFLOW-2005 and transferred in an iterative manner to FLO-2D every time a MODFLOW-2005 time step is reached (Fig. 6). Adapted from Santiago-Collazo et al. (2019)

**37. Figure 9: need to mention in the caption what the insets are and refer to the later figure where they are used.**

**Authors' comment:** The figure required additional editing as it fails to convey meaningful information.

**Action taken**: The figure has been improved as requested and the insets were included (now figures 11 and 12)

[Figure]

**Figure 3.** Distribution of maximum flood depths for Tropical Storm Leslie. The markers indicate repetitive loss properties caused by Tropical Storm Leslie (black), Tropical Storm Andrea (yellow) or other storm events (red). Maximum flood depths at six sample locations (white) are presented in Fig. 11.

[Figure]

Service Layer Credits: Source: Esri, Maxar, GeoEye, Earthstar Geographics, CNES/Airbus DS, USDA, USGS, AeroGRID, IGN, and the GIS User Community

**Figure 4.** Six sample locations (Fig. 10) are selected to observe the maximum flood depths for Tropical Storm Leslie (left). The markers display repetitive loss properties that have been affected by Tropical Storm Leslie (black), Tropical Storm Andrea (yellow), and other storm events (red). The water table timeseries (left) display the behavior of the groundwater heads during Tropical Storm Leslie (blue line), Tropical Storm Andrea (red line) and the 25 May 2020 event (green line) at a specific location (white). Results demonstrate that the simulated water table (right pane) exceeded the surface elevation (brown line) on two locations leading to groundwater-induced flooding (a-b) while the rest are driven by pluvial flooding (c-d-e-f).

**38. Figure 10: This figure has a lot of information and is a little bit hard to read with 1,2,3 and a,b,c showing up multiple times.**

Maybe consider splitting the top part and the bottom part into separate figures. It's not clear what the markers (diamond, triangle, circle) represent here.

Similarly, the VGA Image and Area of Interest markers in the legend are confusing (and maybe not needed). The legend and associated text in the water table plots are way too small and impossible to read. Finally, what is meant with "other events"?

I assume that relates to flood claims from events that were not Leslie or Andrea? At this point the reader has no clue about this information being even shown in the figure and it's not mentioned in the text or caption at all; it's only mentioned later when talking about Fig. 11, so maybe it shouldn't be shown in Fig. 10 to keep the reader focused on what matters.

**Authors' comment:** We thank Reviewer #1 for the suggestion. The figure required major improvement as it failed to convey meaningful information.

**Action taken**: Two figures were created from Figure 10 (now Figures 11-12) as requested to only display relevant markers pertinent to the results section. The readability of both figures has now been improved.

**39. Figure 11: why is the color yellow mentioned in the caption, are the black diamonds and red circles not representing SRL info? Please make sure that figures are 100% understandable when looking at them and reading the caption (one should not have to read the main text to understand the content of a figure).**

**Authors' comment:** We thank Reviewer #1 for the suggestion.

**Action taken**: Figure 11 was removed from the manuscript as it failed to convey meaningful information. All Figures and captions were improved throughout the manuscript as requested.

---

## Author Comment (AC2)

**Response to Anonymous Referee 2**

**General comments:**

**2.1. This manuscript develops an integrated modeling framework to simulate urban flooding caused by rainfall, tides, and groundwater using MODFLOW and FLO-2D. While the idea to combine a surface water model and a groundwater model is intriguing for flooding research, this manuscript suffers many writing and technical deficits.**

> A. **The methodology is not well written and is vague. Details about coupling the FLO-2D model and MODFLOW-2005 model are not well explained. Also, Model setups, including boundary conditions and parameters, are not clear.**
>
> B. **Although model calibration is mentioned in the Abstract, I don't find descriptions of model calibration in the methodology, data, and results. If the model was not calibrated, then the results and analyses would not be convincing.**
>
> C. **The title has the phrase "rainfall-groundwater interaction", but the interactions are not discussed in the manuscript.**
>
> D. **The manuscript fails to provide compelling evidence of interactions among rainfall, tides, and groundwater in the study area. Based on my reading, it seems that the rainfall and tides do not have strong interactions with groundwater in urban areas due to the imperviousness of pavements. High water tables in the study area may be caused by flow from other areas. If the interactions are not significant, then the integrated modeling framework is not useful to the study area.**
>
> E. **Most figures have very poor readability. Some figures are too busy and confusing (e.g. Figures 1-3, 10-12); and some figures fail to convey meaningful information. Please see the specific comments for details.**
>
> F. **The writing is redundant and irrelevant in many places. For example, it is not necessary to provide detailed information about the well-known models (MODFLOW and FLO-2D) in methodology section. Also, most of discussions are irrelevant to the modeling results. Please see the specific comments for details.**

**Based on these serious issues, the manuscript deserves a significant revision. I would not recommend to accept this manuscript for publication.**

*Authors' actions and comments:*
We thank Reviewer #2 for pointing out several constructive criticisms of this manuscript. We proposed a revised version of the manuscript that cleans all irrelevant and unnecessary text, we clarified the confusing parts, specifically for methodology and discussion sections, and we integrated and adjusted the manuscript to address major concerns and remarks by Reviewer #2. To our view, the paper presents novel findings for the following reasons:

1. Several methodologies have been applied to simulate the influence of the water table in lowland watersheds characterized by porous permeable soil. Nevertheless, a large portion of these methodologies are based on qualitative and/or statistical approaches. In recent years

numerical models have slowly been incorporated into groundwater studies, with only a few that use physically-based models to account surface-subsurface water interactions (Yu et al. 2019, Yang & Tsai 2020, Su et al. 2020). This manuscript aims to address this research gap by presenting a loosely-coupled methodology that links two numerical models (FLO-2D and MODFLOW-2005) to simulate surface and subsurface hydrology, producing reasonable results. Here we demonstrate the capabilities of an integrated modelling framework with the potential to simulate compound flooding events.

2. The nature of this paper is heavily linked to a numerical modelling scope, as the manuscript tries to highlight the value of the water table as a key flood driver that can potentially trigger groundwater-induced flooding, with the potential to exacerbate flood conditions. The proposed modelling framework advances understanding on flood modelling at regions with complex urban settings like the Arch Creek Basin or Miami-Dade County, characterized by specific topographic and hydrogeology (flat terrain, porous soil, unconfined aquifers) and subject to surface-subsurface water interactions. Regions prone to groundwater-induced flooding should consider the influence of the water tables in their vulnerability analyses to simulate the influence of groundwater-induced flooding.

3. We developed a simple groundwater model that is based on local modelling efforts (Hughes & White, 2016, Sukop et al. 2018). For simplicity, we approximated the aquifer to be a 2D in the horizontal axis and 1D in the vertical axis. Considering that most of the water table interactions occur in the first aquifer layer of the regional model ($\approx 7$ meters) and the short simulation time of the selected events (64 and 84 hours), we presume that differences in the modelling set up compared to the regional model will not be significant for the purpose of this study. Future work should explore the use of multiple aquifers to assess the differences in the water table dynamics.

4. Despite the lack of quantitative evidence (water level observations), the model was calibrated by using an official dataset that displays properties that have experienced repetitive flood losses across the study area. Figure 11-12 display reasonable results where all spot observations fall within the simulated flood depths. In addition, Figure 13 provides visual evidence (VGI imagery) of flooded conditions for Tropical Storm Andrea. The water table plots show that rainfall-induced flooding is responsible for the flooding conditions in the selected locations as the water table did not exceed the surface elevation.

**Action taken**: Please find our responses to each comment below including how we plan (following the NHESS review process) to adjust the manuscript. We made our best to address major remarks (A to G) as explained in the following points:

***A) The methodology is not well written and is vague. Details about coupling the FLO-2D model and MODFLOW-2005 model are not well explained. Also, Model setups, including boundary conditions and parameters, are not clear.***

We agree with Reviewer #2 that the methodology section contained redundant information about both modelling frameworks and the coupling methodology was not properly explained and requires editing.

**Action taken**: The methodology section has been fully restructured as requested (lines 208 to 369).

[revised manuscript text omitted]

**B) Although model calibration is mentioned in the Abstract, I don't find descriptions of model calibration in the methodology, data, and results. If the model was not calibrated, then the results and analyses would not be convincing.**

Although the manuscript does not include rigorous calibration that is usually required when testing a novel or applied modelling methodology due to the lack of observation data, a calibration approach based on an official property database that have experienced repetitive flooding losses is compared with the simulated flooding conditions. The properties fall within the simulated flood inundation extent. In the context of this research, we demonstrate that surface-subsurface water interactions are localized in the Arch Creek Basin.

In addition, VGI imagery from Tropical Storm Andrea (Fig 12) were used to validate the water table plots with the simulated flood depths, producing reasonable results (Table 3).

Information regarding the model calibration has been inserted in the following sections:

**Abstract (lines 25-26)**
Due to limitations in water level observations, the model was calibrated based on properties that have experienced repetitive flooding losses, and validated using image-based volunteer geographic information (VGI).

**Introduction (lines 104-106)**
Finally, the coupled model results were calibrated based on official database from FEMA, and validated using volunteered geographic information (VGI) flood observations from the study area

**Data Description (line 199-204)**
FEMA's severe repetitive loss properties program is designed to provide grants and financial assistance to residential properties that have experienced frequent flood losses over the years (FEMA, 2021). Currently, seventy-five properties have requested financial assistance for property acquisition or to recoup with some of their investments due to flood damages in the Arch Creek Basin (Miami-Dade, 2017). The database stores detailed information on the date of loss, building type, flood zone designation, type of insurance and claim payments between 1995 to 2015, providing a clear footprint of flooding risk hotspots and flood prone communities. This dataset will be used to calibrate the flood inundation maps.

**Results (line 384-395)**
Simulating surface-subsurface water physical processes through physics-based flood modelling frameworks is relevant and meaningful to better assess the severity of groundwater-induced flooding in low elevation coastal environments characterized by porous permeable soil. Fig. 9 illustrates the simulated maximum inundation depths corresponding to the magnitudes of Tropical Storm Leslie, Tropical Storm Andrea, and the 25 May 2020 storm. Tide levels per se do not pose significant threats to infrastructure as the coastal waters remain within the channels. Fig. 10 illustrates the emergence of the groundwater heads to the surface as a result of the increase in the water table. The simulation proves reasonable in terms of maximum flood depth and extent due to the similarities in the hydrologic conditions, being Tropical Storm Leslie the most severe of all three storms. FEMA's records on properties subject to frequent flooding were used as a calibration approach to verify a match between the model results with flood observations. Although the available records do not specify the observed inundation depths, an agreement between the property locations and maximum water levels may offer sufficient evidence that the model provides reasonable results

(Fig. 11). The calibrated results and display of the water table timeseries in selected locations for Tropical Storm Leslie are shown in Fig. 12-13.

**Conclusions (line 490-494)**
Considering Miami's hydrogeomorphology is one of the most complex globally, the compounding effects of flood drivers may respond differently in diverse geographic settings. Therefore, further research should consider the proposed modeling framework to assess the CF risk in different geographical regions prone to multiple flood drivers, specifically in areas that have access to post-event flooding maps in the form of remote sensing products or VGI data for calibration and validation purposes.

> ***C) The title has the phrase "rainfall-groundwater interaction", but the interactions are not discussed in the manuscript.***

We agree with Reviewer #2 that the title of the paper may cause confusion because groundwater-induced flooding conditions are localized in low-elevation areas within the Arch Creek Basin. For this reason, the manuscript title and the narrative have been modified throughout the document. Now the manuscript title reads "Compound flood modelling framework for surface-subsurface water interactions"

> ***D) The manuscript fails to provide compelling evidence of interactions among rainfall, tides, and groundwater in the study area. Based on my reading, it seems that the rainfall and tides do not have strong interactions with groundwater in urban areas due to the imperviousness of pavements. High water tables in the study area may be caused by flow from other areas. If the interactions are not significant, then the integrated modeling framework is not useful to the study area.***

We really appreciate this concern raised by Reviewer #2.
As mentioned in previous comment 2.1C, the narrative has been improved to clarify the main contribution of this manuscript, presenting the coupling framework between FLO-2D and MODFLOW-2005 to simulate surface-subsurface water processes in localized areas where the water table levels exceed the surface elevation. Thus, we demonstrate that the programming and exchange of information is effective, overcoming important programming obstacles, such as considerable operability differences in their respective numerical codes (lines 276-311)

The three selected events are characterized by heavy precipitation, regular tide levels and unusually high water tables. The storm surge levels do not cause coastal flooding and only increase the water levels on the coast and waterways, having a minimal impact on the flooded area. Nevertheless, the coastal boundary conditions influence the groundwater levels, as the tidal signal can be observed in locations near the coast (Figure 12 b-c-d-e) (lines 408-412).

"The groundwater plots illustrate the effect of tidal and groundwater boundary conditions on the behavior of the simulated water table, in turn demonstrating the importance of both variables in the modeling set-up and influence in subsurface dynamics, as a cyclic high-low pattern characterizes the tide fluctuations of the Biscayne Bay (Fig. 12b – 12e) compared to the defined water heads behavior from well G-852 in the western boundary of the domain (Fig. 12a, 12f). In terms of residential damage, Tropical Storm Leslie and Tropical Storm Andrea may be considered the costliest events in

the Arch Creek Basin as both account for 60% of the reported claims (25 and 17 respectively) (Table 2)."

Although it is possible that the highwater tables in the study area may be caused by flow from other areas, the response of the water table to rainfall events is clearly displayed in Figure 8b-c. Many gauge stations nearby the study site and in Southeast Florida report the same behavior.

The demonstration of significant surface-subsurface water interactions is also provided in Figure 10

[Figure]

Service Layer Credits: Source: Esri, Maxar, GeoEye, Earthstar Geographics, CNES/Airbus DS, USDA, USGS, AeroGRID, IGN, and the GIS User Community

**Figure 2.** Spatial distribution of groundwater-induced flooding for Tropical Storm Leslie (a), Tropical Storm Andrea (b), and 25 May 2020 event (c).

**(line 388 – 389)**
Fig. 10 illustrates the emergence of the groundwater heads to the surface as a result of the increase in the water table.

**And (line 398-402)**
The groundwater flood maps for Tropical Storm Leslie (37.17%), Tropical Storm Andrea (13.87%) and the May 2020 event (20.82%) are showed in Fig. 10. The simulation demonstrates that slight variations in the water table depth (Fig. 8) can exacerbate groundwater emergence extent, resulting in ≈ 10 cm across the Arch Creek Basin. Interestingly heavy precipitations scenarios with very high water tables over extended periods of time (May 2020 event) are more likely to trigger groundwater induced flooding compared to very high precipitation with high water table levels (Tropical Storm Andrea).

> *E)   Most figures have very poor readability. Some figures are too busy and confusing (e.g. Figures 1-3, 10-12); and some figures fail to convey meaningful information. Please see the specific comments for details.*

We thank Reviewer #2 for pointing out this concern. We acknowledge that Figures 1-3, 10-12 were too busy and must be improved, as well as the captions. For this reason, most of the Figures have been redesigned to improve their readability. The captions have also been improved to provide more details.

[Figure]

**Figure 3.** Location map of the study area. (a) MDC located in Southeast Florida, USA (b) current Everglades water flow from Lake Okeechobee towards the Atlantic Coast and Gulf of Mexico, and (c) land survey from 1870 that illustrates the natural flow direction of the Arch Creek to discharge into the Biscayne Bay prior urbanization (Miami Herald, 2019).

[Figure]

**Figure 4.** Aerial photography that compares historical (1948) and current urbanized environment in the study area. (a) Major civil and drainage works contributed to the rapid urbanization of the Arch Creek Basin; (b) Municipality map, including North Miami, Biscayne Park, North Miami Beach, Miami Shores and Unincorporated Miami-Dade (U.S. Department of Agriculture, 1948).

[Figure]

**Figure 5.** Geographical location of selected data in the study site. (a-b) Topographic map showing the location of the Arch Creek Basin (black polygon), and the distribution of closest gauging stations to the study site (black markers), rainfall grid (red square), and properties that have experience severe repetitive losses due to flooding events (yellow).

[Figure]

**Figure 6.** Distribution of maximum flood depths for Tropical Storm Leslie. The markers indicate repetitive loss properties caused by Tropical Storm Leslie (black), Tropical Storm Andrea (yellow) or other storm events (red). Maximum flood depths at six sample locations (white) are presented in Fig. 11.

[Figure]

**Figure 7.** Six sample locations (Fig. 10) are selected to observe the maximum flood depths for Tropical Storm Leslie (left). The markers display repetitive loss properties that have been affected by Tropical Storm Leslie (black), Tropical Storm Andrea (yellow), and other storm events (red). The water table timeseries (left) display the behavior of the groundwater heads during Tropical Storm Leslie (blue line), Tropical Storm Andrea (red line) and the 25 May 2020 event (green line) at a specific location (white). Results demonstrate that the simulated water table (right pane) exceeded the surface elevation (brown line) on two locations leading to groundwater-induced flooding (a-b) while the rest are driven by pluvial flooding (c-d-e-f).

*F)* *The writing is redundant and irrelevant in many places. For example, it is not necessary to provide detailed information about the well-known models (MODFLOW and FLO-2D) in methodology section. Also, most of discussions are irrelevant to the modeling results. Please see the specific comments for details.*

We thank Reviewer #2 for pointing out this concern. We proposed a revised version of the manuscript that cleans all irrelevant and unnecessary text, and we clarify the confusing parts, specifically for methodology (206-243, 370-379) (see comment 2.1A) and discussion sections (439-467).

"The results of this investigation determined that areas in the Arch Creek Basin below 1.0 meter elevation are potentially vulnerable to groundwater-induced flooding (Fig 10, 12a, 12b). Similar results were obtained by Sukop et al. (2018) who found that precipitation as the main trigger for rainfall-induced and groundwater-induced flooding in elevations below 0.9 meters and 1.5 meters respectively, with tidal fluctuations and sea level rise increasing the shallow water table, contributing to the reduction of the storm drain capacity. The present study also determined that antecedent rainfall events were important in the height of the water table at the start of the rainfall events investigated.

A simple groundwater model was approximated to be 2D in the horizontal axis and 1D in the vertical axis. Considering that most of the water table interactions occurred in the upper aquifer layer of the regional model ($\approx$ 7 meters) and the short simulation time of the selected events (64 and 84 hours), we presume that differences in the modelling set up are not significant compared to the regional model and can be considered adequate for the purpose of this study. Additional work may be necessary for the coupled model to be fully operational as the groundwater model should represent the heterogeneous aquifer system to assess the sensitivity of the water table dynamics.

Seasonal water table fluctuations are expected throughout the year, presenting a higher level frequency during the winter and spring seasons due to climate variability and hydrological forcing (Gurdak et al., 2009; Taylor and Alley, 2001). Nevertheless, as we observed with Tropical Storm Leslie and Tropical Storm Andrea, the potential rise of groundwater levels to the surface during dry season cannot be ruled out since the hydraulically non-restrictive nature of the carbonate strata in MDC allows for rapid infiltration and high recharge rates during heavy precipitation events. The hydrologic forcing input and modeling results suggest that the joint occurrence of a high-intensity short-duration precipitation (> 50 mm peak, 250 mm total) with already high groundwater levels (> 1 meter) result in a CF event. Further research on linking multivariate statistical analysis with coupled hydrodynamic modeling frameworks may prove beneficial to identify thresholds that trigger CF conditions (Couasnon et al., 2018; Jane et al., 2020; Moftakhari et al., 2019; Saksena et al., 2019; Sebastian et al., 2017; Serafin et al., 2019).

Although this investigation determined that rainfall and tide levels alone did not produce significant flooding, the modeling efforts did not include storm surge flooding that are often accompany by large hurricanes (Zhang et al., 2013). Nonetheless induced storm surge flooding conditions and sea level rise projections are beyond the scope of this study, future work on assessing the impact of high tide and storm surge induced flooding are fundamental to assess CF events and future flood risk scenarios (Obeysekera et al., 2019)."

*G) **Based on these serious issues, the manuscript deserves a significant revision. I would not recommend to accept this manuscript for publication.***

We really appreciate this concern raised by Reviewer #2. All previous comments have been really helpful to improve the quality and narrative of the manuscript.

As a result, we elaborate on the importance of developing a model capable to simulate surface-subsurface water interactions (Introduction, Lines 95-109), clarify the coupled modeling framework and mathematical compatibility between models (Methodology, 206-311), model configuration (Methodology, 313-368), flood events (371-379), results (381-435) and discussion (439-467).

Replies to the specific comments are reported below.

**Specific comments:**

**2.2. Line 23: I don't find any methodology and results about model calibration throughout the manuscript.**

**Authors' comment:** We agreed with Reviewer #2 that the original version of the manuscript did not properly explain the calibration procedure.

**Action taken**: This comment has been previously addressed on comment 2.1B

**2.3. Line 85: Please spell out the full name of MDC.**

**Authors' comment:** We agree with Reviewer #2 that MDC needs to be spell out and requires editing.

**Action taken**: The sentence has been restructured as requested (Line 87)

**2.4. Line 129: Where is Miami? In section 2.2, it would be better to focus on the rainfall around the study area rather than in a large scale.**

**Authors' comment:** We agree with Reviewer #2 that section 2.2 was misleading as we were referring to Southeast Florida instead of Miami. The rainfall station at Miami International Airport was used by the Florida Climate Center (FCC) to develop the regional study by Abiy et al. (2019).

**Action taken**: We replace "Southeast Florida" to "Miami" as requested in Section 2.2.

**2.5. Line 190-225: A through introduction to FLO-2D is redundant in a scientific paper. Please condense.**

**Authors' comment:** We agree with Reviewer #2 that fully describing FLO-2D is unnecessary for the purpose of this manuscript.

**Action taken**: Section 4.1 has been shortened as requested (Lines 206-216) to address comment 2.1A.

**2.6. Line 226-263: Again. Please condense the introduction to MODFLOW.**

**Authors' comment:** We agree with Reviewer #2 that fully describing MODFLOW-2005 is unnecessary for the purpose of this manuscript.

**Action taken**: Section 4.2 has been shortened as requested (Lines 217-224) to address comment 2.1A.

**2.7. Line 311-321: Please be specific on how the FLO-2D and MODFLOW-2005 are integrated in the algorithm, which is one of the most important contributions in the manuscript. Currently, the descriptions and figures (Fig.4 and Fig.6) are not clear enough.**

**Authors' comment:** We thank Reviewer #2 for highlighting this point and agree that a further explanation on the coupling methodology is required to bring clarity to the manuscript.

**Action taken**: Section 4.3 has been revised and improved as requested (Lines 225-311) to address comment 2.1A and 2.1F. In addition, Fig 4 has been removed from the manuscript. Fig. 5 and Fig. 7 have been improved, and a new figure with the infiltration methodology diagram (Fig. 6) has been added.

**2.8. Line 357-390: Since this is a modeling study, the manuscript should include more details (statements and figures) about the model set-up. Currently, the boundary conditions and parameters of the model are not clear to readers.**

**Authors' comment:** We agree with Reviewer #2's suggestions that the boundary conditions and parameters are not properly explained and requires editing.

**Action taken**: Section 4.4 has been fully restructured as requested (Lines 313-368)

**2.9. Line 380-385: Based on my knowledge, MODFLOW is not able to directly simulate groundwater flow in karst aquifers. Please justify that the groundwater modeling in this study makes sense.**

**Authors' comment:** We thank Reviewer #2 for pointing out this concern. This statement is not necessarily true for every coastal aquifer. We justified the decision of using MODFLOW-2005 to simulate surface-subsurface water interactions based on the published work by (Hughes & White, 2016) and Sukop et al. (2018) (lines 341-347). The first serves as the regional reference model for the County's strategic planning to evaluate sea-level rise and climate change from a water supply, groundwater modelling and saltwater intrusion monitoring perspective.

To our knowledge, MODFLOW is the trademark and only model use for groundwater modelling simulations in Miami and southeast Florida.

**Action taken**: A new paragraph was included to present the works that apply MODFLOW in the region (lines 343-349).

"Concerning MODFLOW-2005, a simple model was developed based on the regional groundwater model of MDC developed by USGS (Hughes & White, 2016) using an advanced version of MODFLOW-2005 that applies the Newton-Raphson formulation (MODFLOW-NWT) with the Surface-Water Routing (SWR1) Process to simulate comprehensive surface and groundwater hydrologic conditions on a 15 meter grid resolution; the second model consists of a local 1D MODFLOW that simulates the influence of the water table on flooding conditions in an upper portion of the Arch Creek Basin (Sukop et al., 2018). "

**2.10. Line 381: The model is composed of one-layer...**

**Authors' comment:** We agree with Reviewer #2's suggestions that the boundary conditions and parameters were not properly explained and requires editing. We approximated the entire three layer system (Hughes & White, 2016) as a homogenous aquifer (Sukop et al. 2018).

**Action taken**: This comment has been previously addressed on comment 2.1.3 and 2.9 (lines 343-361).

In addition, the discussion section has been strengthened to address this observation (Lines 446-451).

"A simple groundwater model was approximated to be 2D in the horizontal axis and 1D in the vertical axis. Considering that most of the water table interactions occurred in the upper aquifer layer of the regional model ($\approx$ 7 meters) and the short simulation time of the selected events (64 and 84 hours), we presume that differences in the modelling set up are not significant compared to the regional model and can be considered adequate for the purpose of this study. Additional work may be necessary for the coupled model to be fully operational as the groundwater model should represent the heterogeneous aquifer system to assess the sensitivity of the water table dynamics."

**2.11. Line 390: "Groundwater elevations" is not a clear term.**

**Authors' comment:** We agree with Reviewer #2 that the term 'groundwater elevations' is not clear and requires editing.

**Action taken**: The term has been changed to 'groundwater heads' as requested

**2.12. Line 396-416: Redundant. Please condense.**

**Authors' comment:** We agree with Reviewer #2 that section 4.5 is redundant and requires editing.

**Action taken**: Section 4.5 has been fully restructured as requested (Lines 371-379)

**2.13. Line 429-430: Please provide the evidence to support that the water from rainfall and tides rapidly infiltrates. Based on my knowledge, infiltration in urban areas should not be significant. High groundwater table should be a result of flow from other regions.**

**Authors' comment:** We thank Reviewer #2 for highlighting this point. Although we agreed that groundwater systems are sensitive to water table levels from other regions, this statement is not necessarily true for the Arch Creek Basin and MDC. The tidal signal can be observed in the groundwater levels as the tides wary in areas near the coast (Fig. 12b,c,d,e).

Similarly, the Arch Creek Basin is not entirely impervious, and the infiltration happens in most of the basin. For example, infiltration is possible in most housing due to the green cover in the front and back. The water table rises as a result of the rainfall infiltration. Many water table gauges (including the presented rainfall events) show this behavior.

**Action taken**: This comment has been previously addressed on comment 2D.
See satellite image to observe the green cover in the study area.

[Figure]

**2.14. Line 450-458: I can't find that information in Figure 10 and 11.**

**Authors' comment:** We agree with Reviewer #2 that the statements do not match Figures 10 and 11 (now 12).

**Action taken**: The results section has been improved and associated statements to Figure 9 (line 386-388),

[revised manuscript text omitted]

**2.15. Line 466-474: Figure 12 is too hard to read. I can't follow the statements with the figure.**

**Authors' comment:** We agree with Reviewer #2 that figure 12 (now figure 13) is too hard to read and requires major editing.

**Action taken**: Figure 13 and associated statements (line 424-435) have been improved for readability purposes. (see comment 2.14)

[Figure]

**Figure 1.** Maximum flood depths for Tropical Storm Andrea in the Northwestern portion of the Arch Creek Basin (top). The marker (yellow) display properties that were affected during Tropical Storm Andrea. Three sample locations (white) are presented as subdomains (a-b-c) and available crowdsourced observations display the flooding conditions at a specific cell (white). The simulated water table timeseries (right pane) show that groundwater heads remained below the surface elevation (brown line); thus, all three locations experienced rainfall-induced flooding.

**2.16. Line 485-496: The writing is irrelevant to the model results. It might be put in the introduction section.**

**Authors' comment:** We agree with Reviewer #2 that the content is irrelevant to the discussion section.

**Action taken**: Lines 485-496 have been moved to Section 2.4 (flood risk and vulnerability) as it provides relevant background information of the study site (Lines 145-162).

**2.17. Line 498-499: Please provide associating model results to support the statement.**

**Authors' comment:** We thank Reviewer #2 for highlighting this point.

**Action taken**: This comment has been previously addressed on comment 2D and 2.13.

**2.18. Line 514-520: Irrelevant writing.**

**Authors' comment:** We thank Reviewer #2 for the suggestion.

**Action taken**: The irrelevant writing and title subsections have been removed. A more concise discussion is now provided (lines 439-467)

**2.19. Line 528-535: Irrelevant writing.**

**Authors' comment:** We thank Reviewer #2 for the suggestion.

**Action taken**: This comment has been previously addressed on comment 2.18

**2.20. Line 825: Figure 6 is confusing. Please clarify the meaning of T, DT, and dt.**

**Authors' comment:** We thank Reviewer #2 for the suggested correction. The figure requires additional editing. We apologize for the confusion due to the lack of legend.

**Action taken**: This comment has been previously addressed on comment 2.1A

**2.21. Line 835: Groundwater table in Figure 8a is not influenced by rainfall and tides, compared to the other two figures.**

**Please justify that the data in Figure 8a is correct.**

**Authors' comment:** We thank Reviewer #2 for pointing out this concern. The water table time series of Figure 8 is not a mistake. As mentioned in section 3.2.3 (groundwater heads) water table levels were reported on a daily basis before October 2007 and Tropical Storm Leslie (2-4 October 2000)

falls in this category. The lag on the water table response to precipitation is a time resolution issue by the USGS.

**Action taken**: No action was taken.

**2.22. Line 840: Caption of figure 9 is confusing.**

**Authors' comment:** We agree with Reviewer #2 that caption of figure 9 fails to convey meaningful information.

**Action taken**: The figure and caption has been improved for readability purposes. This comment has been previously addressed on comments 1.37 and 2.1E

**2.23. Line 845: The caption of figure 10 does not match the figure. The color represents flood depth but only water surface elevation is mentioned in the caption. Also, only one storm event is mentioned in the caption but many events are included in the figure.**

**Authors' comment:** We thank Reviewer #2 for the suggestion. We agree that figure 10 requires major improvements as it fails to convey meaningful information.

**Action taken**: Two figures were created to improve the readability of both figures and captions. This comment has been previously addressed on comments 1.37 and 2.1E

---

## Referee Report (RR1)

The reviewer thanks for the authors' responses to the comments on the first draft as well as the excellent revisions. The author properly addressed most of my comments and concerns in the response letter. Details about the data and methodology are included in the revised manuscript, and the results/discussions are scientifically sound, which is much better than the first draft.

There is one remaining concern that may be considered by the author. In line 308-309, the term "surface depth" is not clear. If it refers to the surface water depth, then the conditional statement at the lower-right corner of figure 6 might be "$h_{GW} > h_{WSD}$". Other than this, I would recommend to finally accept this manuscript.

---

## Author Response (AR2)

**Response to Anonymous Referee #2**

**1. The reviewer thanks for the authors' responses to the comments on the first draft as well as the excellent revisions. The author properly addressed most of my comments and concerns in the response letter. Details about the data and methodology are included in the revised manuscript, and the results/discussions are scientifically sound, which is much better than the first draft.**

**There is one remaining concern that may be considered by the author. In line 308-309, the term "surface depth" is not clear. If it refers to the surface water depth, then the conditional statement at the lower-right corner of figure 6 might be "hGW > hWSD". Other than this, I would recommend to finally accept this manuscript.**

**Authors' comment:** We thank Reviewer #2 for the positive feedback and the suggested correction.

**Action taken**: Line 308-309 has been restructured as requested: If the groundwater heads calculated in MODFLOW-2005 are higher than the water surface elevation in FLO-2D, the depth of water from groundwater will be added to the water surface depth.

In addition, the term "water surface elevation" was added to Figure 6 to be consistent with the text: